# Semantic-Drive: Trustworthy and Efficient Long-Tail Data Curation via Open-Vocabulary Grounding and Neuro-Symbolic VLM Consensus

**Antonio Guillen-Perez**                                                                 *antonio_algaida@hotmail.com*
*Independent Researcher*
*antonioalgaida.github.io*

**Reviewed on OpenReview:** *https://openreview.net/forum?id=qN2oN36L3k*

## Abstract

The development of Autonomous Vehicles (AVs) is currently hampered by a scarcity of long-tail training data. While fleets collect petabytes of video logs, identifying rare safety-critical events, specifically scenarios like erratic jaywalking or complex construction diversions, remains a manual process that is often cost-prohibitive. Existing automated solutions rely either on coarse metadata search, which lacks semantic precision, or on cloud-based Vision-Language Models (VLMs), which introduce privacy concerns and computational overhead. In this work, we introduce **Semantic-Drive**, a local-first, neuro-symbolic framework designed for verifiable semantic data mining. Our approach decouples perception into two distinct stages: (1) **Symbolic Grounding** via a real-time open-vocabulary detector (YOLOE) to anchor attention, and (2) **Cognitive Analysis**, where a Reasoning VLM performs forensic scene analysis. To reduce hallucinations and reliability issues common in generative models, we implement a "System 2" inference-time alignment strategy that utilizes a multi-model "Judge-Scout" consensus mechanism. When benchmarked on the nuScenes dataset against the Waymo Open Dataset (WOD-E2E) taxonomy, it was observed that Semantic-Drive achieves a **recall of 0.966** on safety-critical scenarios (vs. 0.331 for OWL-v2 and 0.271 for Grounding DINO). Notably, the system reduces risk assessment error by **40%** compared to single-model baselines. The entire pipeline runs on consumer hardware (NVIDIA RTX 3090), offering an accessible and privacy-preserving alternative to cloud-native architectures.

## 1 Introduction

A primary challenge in scaling autonomous perception systems is not only the volume of data collected, but also the *imbalanced distribution* of that data. As illustrated in Figure 1, driving scenarios follow a heavy-tailed (Zipfian) distribution where the vast majority of collected logs, representing approximately 99% of the total dataset, consist of nominal, low-entropy driving conditions such as highway cruising or stopped traffic (Liu & Feng, 2024). While abundant, this "Head" of the distribution offers diminishing returns for improving model performance. The value for Level 4 safety validation lies heavily in the "Long Tail," specifically in rare, high-entropy events such as construction zones with conflicting lane markings, erratic vulnerable road users (VRUs), or sensor degradation due to sudden weather changes (Caesar et al., 2019). Identifying these samples within large-scale data lakes creates a bottleneck in retrieving "dark data." Currently, manual review is cost-prohibitive at this scale. Heuristic metadata tags (e.g., `weather=rain`) lack the semantic granularity to reliably distinguish between a wet road and a hydroplaning risk.

Mining these scenarios from video footage remains a persistent challenge. Traditional methods rely on rigid heuristics, such as querying CAN bus data for hard braking events or searching for metadata keywords, which often suffer from poor temporal granularity. While recent Vision-Language Models (VLMs) like GPT-4V (OpenAI et al., 2023) offer semantic understanding, relying on closed-source cloud APIs for data curation is

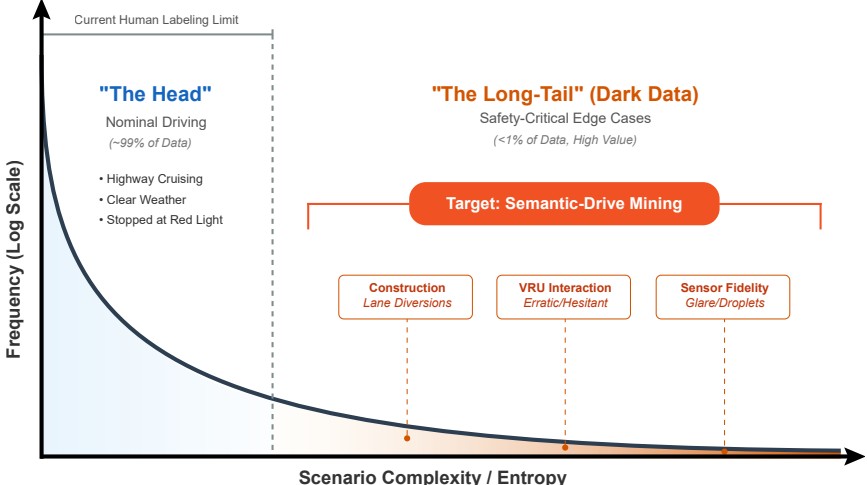

Figure 1: **The "Dark Data" Bottleneck in Autonomous Driving.** The distribution of driving scenarios follows a Power Law (Zipfian) distribution. **(Left) The "Head":** Represents 99% of data logs, consisting of nominal, low-entropy driving (e.g., highway cruising) which provides diminishing returns for model training (Liu & Feng, 2024). **(Right) The "Long Tail":** Contains rare, safety-critical edge cases defined by the Waymo Open Dataset (WOD-E2E) taxonomy, such as erratic VRUs or sensor degradation. Traditional human annotation is cost-prohibitive for mining this region. **Semantic-Drive** automates the retrieval of these high-value samples.

largely impractical for the automotive industry. This limitation stems from data privacy regulations (GDPR (European Parliament & Council of the European Union)), bandwidth constraints, and the computational cost of processing video streams at scale.

To address this gap, we introduce **Semantic-Drive**, a privacy-aware, local-first framework for semantic data mining. Unlike end-to-end driving agents that utilize VLMs for *control* (Xu et al., 2024), Semantic-Drive focuses on *retrieval*. It acts as a "Cognitive Indexer" that transforms unstructured video logs into a queryable semantic database.

Our approach is based on a **Neuro-Symbolic Architecture** designed to run on consumer-grade hardware (e.g., a single NVIDIA RTX 3090). It is documented that pure VLMs often suffer from hallucination and "small-object blindness"; for example, advanced vision-language models tend to perform poorly on fine, small objects and produce inconsistent labels (Sun et al., 2025). To mitigate this issue, our framework separates perception into two distinct pathways: (1) A symbolic grounding stage using real-time open-vocabulary object detection to generate an inventory of hazards, and (2) A cognitive reasoning stage where a Chain-of-Thought (CoT) VLM performs forensic analysis to verify detections and assess risk. This architecture is visualized in Figure 2.

While we primarily evaluate this architecture within the autonomous driving domain, the underlying "Judge-Scout" consensus mechanism addresses a recurring challenge in Trustworthy AI: the reliability of black-box reasoners in high-stakes environments. Prior work has shown that AI systems deployed in safety-critical contexts face risks from unintended failures without structured safeguards for reliability (Amodei et al., 2016), and that relying solely on opaque models can be problematic for decisions where verifiable behavior is essential (Rudin, 2019). The pattern of pairing a high-recall symbolic detector with a skeptical, reasoning-heavy model aligns with research on hybrid neural-symbolic systems that seek to combine robust perception with structured inference (Besold et al., 2017). This pattern is also used outside driving in other domains that require high precision. For example, deep networks have been applied to chest X-ray anomaly detection, where missing a critical finding is highly detrimental and verification is necessary (Rajpurkar et al., 2017). Comprehensive anomaly datasets like MVTec AD are a benchmark for reliability in industrial quality inspection, where

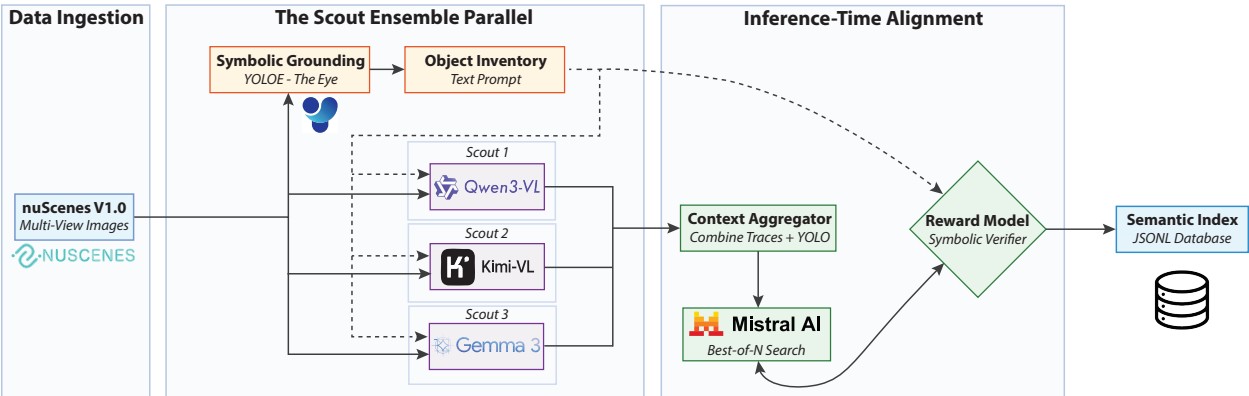

Figure 2: **The Semantic-Drive System Architecture.** The pipeline operates in three stages: (1) **Symbolic Grounding (Orange):** YOLOE extracts a textual object inventory from raw frames. (2) **Cognitive Scouting (Purple):** An ensemble of heterogeneous Reasoning VLMs (Qwen, Kimi, Gemma) performs independent forensic analysis, anchored by the symbolic inventory. (3) **Consensus & Alignment (Green):** The Judge (Ministral-3-14B) synthesizes the scout reports. A deterministic Reward Model performs inference-time verification (*Best-of-N*) to filter hallucinations before committing to the final Semantic Index.

defects are rare but costly to miss (Bergmann et al., 2019). In these settings, as in driving, the cost of a false negative is severe, yet the compute budget for manual review is strictly limited.

Our contributions are as follows:

- **Neuro-Symbolic "System 2" Architecture[1]:** We introduce an inference-time alignment strategy inspired by System 2 reasoning (Bengio, 2019). By enforcing a "Skepticism Policy" where the VLM must logically verify symbolic detections against visual evidence, we measurably reduce hallucinations compared to standard zero-shot prompting.

- **Judge-Scout Consensus Mechanism:** We address the stochastic nature of LLMs through a multi-model consensus engine. We demonstrate that aggregating reasoning traces from heterogeneous scouts (e.g., Qwen and Gemma) via a "Judge" improves risk assessment reliability.

- **Mapping the WOD-E2E Taxonomy:** We convert the Waymo Open Dataset (WOD-E2E) taxonomy (Xu et al., 2025) into a structured JSON schema. This allows us to automatically extract detailed causal attributes, such as "Implicit Lane Diversion" and "Sensor Fidelity Issues," which global embeddings frequently overlook.

- **Accessible Data Curation:** We show that scenario curation can be done on consumer hardware. Our local pipeline reduces the marginal cost of curation compared to cloud-based solutions, making automated data mining accessible without relying on external services.

It is important to clarify the operational role of Semantic-Drive within the autonomous driving ecosystem. While recent end-to-end architectures focus on real-time onboard control and trajectory planning (e.g., UniAD (Hu et al., 2022)), Semantic-Drive is designed as an **offline DataOps engine**. It is intended to transform unlabeled data lakes into structured, queryable semantic indices. By automating the discovery of rare, safety-critical scenarios, our framework enables engineers to curate the specific long-tail datasets required to improve and validate downstream real-time planners. In this workflow, Semantic-Drive acts as a synergistic precursor to end-to-end stacks, transferring the retrieval burden from manual data review to logic-driven semantic search.

---

[1]In dual-process theory, this denotes slow, deliberate, and logical reasoning, contrasted with fast, intuitive System 1 processing (Kahneman, 2011).

**Reproducibility and Open Science.** To support future research and ensure reproducibility of our pipeline, we release all artifacts associated with this project. The complete source code and evaluation scripts are available in our GitHub repository[2]. To facilitate reproducibility, we release the full generated semantic index ($N = 2,550$) and the annotated Gold Set via Hugging Face Datasets[3]. Additionally, we host an interactive "Data Explorer" visualizing the mined scenarios on Hugging Face Spaces[4].

## 2 Related Work

### 2.1 Vision-Language Models in Autonomous Driving

The integration of Large Multimodal Models (LMMs) into the autonomous driving stack has largely been divided into two streams: generative simulation and end-to-end control. Systems such as DriveGPT4 (Xu et al., 2024), DriveVLM (Tian et al., 2024), and Waymo's EMMA (Hwang et al., 2024) leverage VLMs to map sensor inputs directly to control actions or to generate natural language explanations for driving decisions. These approaches demonstrate strong reasoning capabilities but often require substantial computational resources. As noted by Kondapally et al. (2026), current VLMs designed for clear-weather scenarios often rely on expensive backbones, leading to high memory usage and slow inference speeds that hinder real-time decision-making in transitional environments.

Furthermore, relying on closed-source or cloud-hosted models for driving tasks introduces trustworthiness concerns. The recent AutoTrust benchmark (Xing et al., 2024) revealed that specialized DriveVLMs are vulnerable to disclosing sensitive information, raising privacy questions for fleets operating in public spaces. Semantic-Drive complements this ecosystem by shifting the focus from control to curation. Rather than acting as a real-time agent, our framework serves as a foundational data operations engine. By operating locally on consumer hardware, we address the privacy and latency bottlenecks identified in AutoTrust, facilitating the discovery of long-tail training samples to improve downstream driving agents.

### 2.2 Semantic Data Mining and Scenario Retrieval

Retrieving safety-critical edge cases from unlabeled data remains an active area of research. Current methodologies typically fall into three broad categories:

- **Heuristic Metadata Search:** Standard datasets like nuScenes (Caesar et al., 2019) rely on manual tags or CAN-bus triggers such as hard braking. While recent studies by Guillen-Perez (2025) demonstrated that uncertainty-based weighting can improve offline RL performance, these methods still largely rely on kinetic proxies rather than true semantic understanding. As shown in our experiments, this approach suffers from poor temporal granularity by flagging entire scenes based on transient events.

- **Programmatic Mining:** Approaches like RefAV (Davidson et al., 2025) synthesize programmatic queries (e.g., SQL or Python) to filter track-level data. While computationally efficient, they are constrained by geometric primitives. These systems often miss visual nuances such as construction signage, debris types, or pedestrian gaze direction.

- **Latent Embedding Search:** Methods like VLMine (Ye et al., 2024) and localized CLIP searches utilize vector similarity to find scenarios. However, global embeddings suffer from "bag-of-words" blindness. They often fail to distinguish between a pedestrian on the sidewalk (safe) and one in the lane (hazard) due to a lack of precise spatial binding (Zhong et al., 2021).

In contrast, Semantic-Drive introduces a causal reasoning layer. Instead of relying on geometric tracks or statistical keyword frequency, we employ Chain-of-Thought (CoT) reasoning to analyze the implications of a scene. This strategy mirrors recent advances in fine-grained video reasoning, such as VideoEspresso (Han

---

[2]https://github.com/AntonioAlgaida/Semantic-Drive
[3]https://huggingface.co/datasets/agnprz/semantic-drive-results
[4]https://huggingface.co/spaces/agnprz/Semantic-Drive-Explorer

et al., 2024), which utilizes CoT annotations and multiple foundation models to extract logical relationships from sparse core frames. While using multiple models for automated annotation is an established paradigm, Semantic-Drive differentiates itself by enforcing a strict neuro-symbolic hierarchy. Rather than relying purely on LLM-based consensus, our reasoning ensemble is explicitly constrained by a deterministic object inventory and a rule-based reward function. By contextualizing spatial evidence through this guided multi-model reasoning, we enable the retrieval of scenarios defined by complex causal interactions that are often missed by geometric or latent-search methods.

### 2.3 Neuro-Symbolic Grounding and Hallucination Mitigation

A primary barrier to deploying VLMs in safety-critical domains is hallucination, specifically the tendency to invent objects or misinterpret spatial relationships. To mitigate this issue, recent research has explored neuro-symbolic architectures that pair a symbolic detector with a cognitive reasoner. This aligns with the "System 1" intuitive processing versus "System 2" deliberate reasoning distinction in cognitive science (Kahneman, 2011), a framework that has been proposed as a roadmap for the next generation of deep learning (Bengio, 2019).

Recent work in visual reasoning has demonstrated that decomposing complex tasks into closed-loop modification steps can improve performance. For instance, Liu et al. (2025) showed that autonomous imagination enables MLLMs to solve tasks that initially exceed their perceptual capacity. Similarly, our framework enforces a "System 2" skepticism policy. We utilize the output of a high-recall open-vocabulary segmentor (YOLOE (Wang et al., 2025)) not as the final result, but as a grounded attention anchor for the VLM. Unlike zPROD (Sinhamahapatra et al., 2025), which uses VLMs primarily for bounding box refinement, our framework requires the VLM to logically verify the symbolic inventory against visual evidence. This bidirectional validation functions similarly to tool-augmented generation in natural language processing, where the model uses an external tool to fact-check its internal representations, reducing false positives compared to pure neural approaches.

## 3 Methodology

The Semantic-Drive framework functions as a local-first data curation pipeline. Unlike cloud-based solutions that process video streams via external APIs, our architecture is designed to operate entirely on consumer-grade hardware, specifically a single NVIDIA RTX 3090 with 24 GB VRAM. The system processes raw, synchronized multi-camera feeds into a structured semantic database through a four-stage neuro-symbolic process: (1) Symbolic Grounding, (2) Cognitive Analysis, (3) Multi-Model Consensus, and (4) Inference-Time Alignment.

### 3.1 Stage 1: Symbolic Grounding (The Eye)

To address small-object detection failures and spatial hallucinations frequently observed in pure Vision-Language Models (VLMs) (Sun et al., 2025), we incorporate a symbolic prior. We utilize **YOLOE** (Wang et al., 2025), a real-time open-vocabulary segmentation model, to perform an initial sweep of the visual field. Instead of standard COCO classes, we inject a custom text-prompt taxonomy aligned with the Waymo Open Dataset for End-to-End Driving (WOD-E2E) (Xu et al., 2025). This inventory targets specific long-tail categories such as `orange drum`, `jersey barrier`, `debris`, and `puddle`.

The detector output is not saved directly as a final prediction. Instead, it is converted into a structured textual *Object Inventory* that serves as a prompt injection for the downstream VLM. Formally, we define the inventory set $\mathcal{I}$ as:

$$\mathcal{I} = \{(Class_i, Cam_j, c_i, S_{rel}) \mid c_i > \tau_{recall}\} \tag{1}$$

where $Class_i$ represents the semantic category, $Cam_j$ denotes the spatial origin (Front-Left, Center, or Right), $c_i$ is the model confidence score, and $S_{rel}$ measures the relative size of the bounding box. We empirically calibrated a low-confidence threshold ($\tau_{recall} = 0.15$) to prioritize recall over precision at this initial stage. This design choice deliberately admits potential false positives, such as reflections on wet roads, into the context window, delegating the burden of rejection to the subsequent reasoning module.

### 3.2 Stage 2: Cognitive Analysis (The Brain)

The central component of our framework is the reasoning module, where a quantized VLM (e.g., Qwen3-VL-30B) performs forensic scene analysis. We employ a **Front-Hemisphere Attention Strategy** by providing synchronized Front-Left, Front-Center, and Front-Right images to maximize spatial resolution within the model context window.

The VLM receives both the raw pixel data and the symbolic *Object Inventory*. We enforce a **"Skepticism Policy"** via the system prompt to handle the low-confidence detections generated in Stage 1. For high-confidence detections ($c > 0.8$), the VLM is instructed to accept the object's existence while verifying its semantic context (e.g., whether a pedestrian is interacting with the roadway). For low-confidence detections ($c < 0.5$), the VLM must treat the detection as a hypothesis and perform a visual verification step to reject artifacts.

### 3.3 Stage 3: Multi-Model Consensus (The Judge)

Individual VLMs exhibit stochastic behavior and inductive biases. To improve curation reliability without human intervention, we introduce a **"Judge-Scout" Architecture**. We deploy $N$ heterogeneous "Scout" models (e.g., Qwen3-VL (Bai et al., 2025), Kimi-VL (Team et al., 2025b), Gemma-3 (Team et al., 2025a)) to process the same frame in parallel. A separate "Judge" LLM (Ministral-3-14B (Liu et al., 2026)) aggregates their JSON outputs and reasoning traces. The Judge applies a **Safety-Bias Voting Logic**: if Scouts disagree on a high-risk attribute, the Judge favors the positive detection, provided valid reasoning is documented in the trace.

### 3.4 Stage 4: Inference-Time Alignment (Symbolic Verification)

To mitigate stochastic hallucinations without requiring resource-intensive fine-tuning, we implement an inference-time search strategy inspired by *Best-of-N* sampling (Gui et al., 2024). The Consensus Judge generates $N = 3$ candidate scenario descriptions. We rank these candidates using a deterministic, rule-based reward function $R(y, \mathcal{I}_{sym})$ that penalizes outputs not corroborated by the visual evidence:

$$R(y) = \alpha \cdot \mathbb{I}_{grounding} + \beta \cdot \mathbb{I}_{causality} - \gamma \cdot \mathbb{I}_{hallucination} \tag{2}$$

The system outputs the candidate $y^* = \operatorname{argmax}_y R(y)$. To clarify the components of Equation 2, we define the terms as explicit indicator functions:

- $\mathbb{I}_{grounding} \in \{0, 1\}$ equals 1 if the safety-critical tags generated by the VLM map directly to objects present in the Stage 1 symbolic inventory.

- $\mathbb{I}_{causality} \in \{0, 1\}$ equals 1 if the VLM logically connects an identified hazard to a specific required ego-vehicle maneuver.

- $\mathbb{I}_{hallucination} \in \{0, 1\}$ equals 1 if the VLM describes a critical hazard that is absent from both the symbolic inventory and the aggregated scout reports.

The weights serve as a symbolic gatekeeper for the reasoning engine. We determined these parameters via a grid search to prioritize safety and grounding, resulting in $\alpha = 2.0$ (grounding weight), $\beta = 1.5$ (causal logic), and $\gamma = 10.0$ (hallucination penalty). The comparatively high value for $\gamma$ enforces the skepticism policy, ensuring that agents reported by the VLM must be grounded in the symbolic Stage 1 inventory to be included in the final semantic index. A detailed sensitivity analysis of these hyperparameters is provided in Appendix C, demonstrating how they control the trade-off between precision and recall.

### 3.5 Theoretical Analysis

#### 3.5.1 Computational Complexity of Consensus

Let $C_{scout}$ be the inference cost of a single VLM scout and $C_{judge}$ be the cost of the aggregation step. The total computational cost for a consensus mechanism with $K$ scouts is given by:

$$C_{total} = \sum_{k=1}^{K} C_{scout}^{(k)} + C_{judge} \tag{3}$$

Since the scouts operate independently, the complexity scales linearly as $O(K)$. While this increases the latency per frame compared to a single model ($C_{total} \approx K \times C_{single}$), our empirical results in Section 5.4 demonstrate that the risk assessment error decreases consistently as the ensemble size grows. This behavior aligns with the Condorcet Jury Theorem, which establishes that the probability of a correct majority decision increases with the number of independent voters, provided individual accuracy exceeds random chance. For offline data curation where throughput is secondary to labeling precision, this computational trade-off is justified.

#### 3.5.2 Safety Constraints and the Penalty Term

We frame the selection of the hallucination penalty ($\gamma = 10.0$) in our reward function as a constrained optimization problem rather than relying on an arbitrary hyperparameter. Ideally, we wish to maximize the semantic richness $S(y)$ of the description, subject to the strict constraint that the Hallucination Rate $H(y) = 0$.

$$\max_{y} S(y) \quad \text{s.t.} \quad H(y) = 0 \tag{4}$$

By converting this to an unconstrained objective $S(y) - \lambda H(y)$, the parameter $\gamma$ functions as the Lagrange multiplier $\lambda$. Setting $\gamma \to \infty$ effectively enforces a zero-tolerance policy for hallucination. We empirically set $\gamma = 10.0$ as a soft approximation. This specific configuration allows the model to override the symbolic detector only in cases where the causal reasoning evidence ($\beta$) is highly substantial, thereby balancing safety constraints with the ability to correct symbolic false negatives.

## 4 The "Scenario DNA" Taxonomy

Standard autonomous driving datasets predominantly rely on flat metadata tags such as `rain=True` or `pedestrian=True`. Although useful for coarse filtering, these binary indicators often fail to capture the causal dynamics required for validating Level 4 systems. For example, a pedestrian on a sidewalk represents a nominal event, whereas a pedestrian hesitating at the curb during a rainstorm constitutes a high-value edge case for training.

To address this limitation, Semantic-Drive extracts a hierarchical "Scenario DNA" structure, which is illustrated in Figure 3. We define a comprehensive ontology designed to capture the interaction between environmental constraints, static topology, and dynamic agent intent. This schema is strictly typed using enumerations to ensure database normalization and to align with the WOD-E2E taxonomy (Xu et al., 2025).

### 4.1 Layer 1: ODD and Phenomenology

This layer characterizes the phenomenological constraints of the scene and serves as a primary filter for perception teams during data mining. Unlike standard weather tags, we explicitly model sensor integrity:

- **Environmental Conditions:** Fine-grained distinctions such as `heavy_rain` versus `mist` that significantly affect sensor range.

- **Sensor Fidelity:** A targeted feature for scenario mining. We detect localized failure modes such as `lens_flare`, `droplets_on_lens`, and `motion_blur`. Identifying these frames allows teams to build robust de-hazing datasets.

**Semantic-Drive "Scenario DNA"**

| 1. ODD & Physics | 2. Map Topology | 3. Agent Dynamics | 4. Causal Logic |
|---|---|---|---|
| • Weather (Rain/Fog)
• Time of Day
• Lighting (Glare)
• Surface Friction
• **Sensor Integrity (Droplets/Flare)** | • Scene Type
• Lane Config
• Traffic Controls
• **Map Divergence (*Drivable Area Status*)** | • VRU Status
• Lead Vehicle
• Adjacent Vehicle
• **Intent/Behavior (*Hesitant/Aggressive*)** | • Primary Challenge
• Blocking Factor
• **Ego-Action (*Nudge/Yield/Stop*)**

**Risk: 0-10** |

Figure 3: **The "Scenario DNA" Hierarchical Taxonomy.** Unlike standard flat metadata tags, Semantic-Drive enforces a causal dependency chain. **(1) ODD and Physics:** Defines environmental constraints, including specific attributes like sensor integrity (e.g., `lens_flare`). **(2) Map Topology:** Identifies contradictions between the high-definition map and reality (Map Divergence). **(3) Agent Dynamics:** Infers behavioral intent (e.g., hesitation) rather than just presence. **(4) Causal Logic:** Synthesizes the previous layers into a planner-centric risk assessment.

## 4.2 Layer 2: Topology and Map Divergence

Level 4 autonomous systems rely heavily on high-definition maps. An important failure mode occurs when the physical world contradicts the preloaded map, commonly referred to as *map divergence*. Our schema targets these anomalies:

- **Drivable Area Status:** We categorize obstructions into `restricted_by_static_obstacle`, such as construction cones, or `physically_restricted`, such as floodwaters.

- **Lane Configuration:** We identify temporary shifts such as `lane_diversion` or `merge_left`. These are common in construction zones but are often absent from static high-definition maps.

## 4.3 Layer 3: Actor Dynamics and Intent

While traditional object detection provides bounding boxes representing physical presence, Semantic-Drive infers behavioral intent:

- **VRU Status:** We differentiate between `roadside_static`, which implies low risk, and `jaywalking_hesitant`, which implies higher prediction uncertainty. The latter is useful for training prediction models to handle ambiguity.

- **Vehicle Dynamics:** We detect aggressive behaviors such as `cutting_in`, `tailgating`, or `drifting` by leveraging the temporal context implied by vehicle pose and road placement.

## 4.4 Layer 4: Causal Criticality (The Planner Layer)

Finally, we synthesize the preceding layers into a planner-centric assessment. This layer identifies the causal etiology of the scenario difficulty:

- **Primary Challenge:** We classify the root cause of difficulty, such as `occlusion_risk` at a blind corner, `prediction_uncertainty` regarding an erratic agent, or `violation_of_map_topology`.

- **Ego-Maneuver:** We identify the implied necessary action for safety, such as `nudge_around_obstacle` or `unprotected_turn`. This allows planning engineers to query specifically for scenarios requiring complex maneuvers.

# 5 Experiments and Evaluation

## 5.1 Experimental Setup

To evaluate the framework across diverse environments, we deployed Semantic-Drive on the full `nuScenes v1.0-trainval` dataset (Caesar et al., 2019), which comprises 850 distinct driving scenes collected in Boston and Singapore. We extracted synchronized Front-Left, Front-Center, and Front-Right camera feeds for every processed keyframe.

**Hardware Infrastructure:** All inference, including VLM scouting and LLM judging, was conducted locally on a consumer-grade workstation equipped with a single **NVIDIA RTX 3090 (24 GB VRAM)**. This constraint tests the efficiency of our framework, demonstrating that data curation can be performed without large-scale cloud computing resources.

**Data Sampling Strategy:** To evaluate the system across the full diversity of the dataset within a feasible compute budget, we employed a scene-level sparse sampling strategy. We extracted $k = 3$ keyframes per scene (Start, Middle, End), resulting in a curated dataset of 2,550 unique semantic fingerprints. This approach ensures that every specific environmental context and geographic location in the nuScenes validation set is represented.

**Model Configuration:**

- **Symbolic Grounding:** YOLOE-11L-Seg (FP16) configured with a custom WOD-E2E open-vocabulary taxonomy.

- **Cognitive Scouts:** We deployed a heterogeneous ensemble of models: **Qwen3-VL-30B-Thinking**, **Kimi-VL-Thinking**, and **Gemma-3-27B-IT**. These models were deployed using 4-bit quantization (`Q4_K_M`).

- **Consensus Judge:** We utilized **Ministral-3-14B-Instruct** as the local decision engine, which was selected for its architectural differences from the scouts to help mitigate shared inductive biases.

## 5.2 Ground Truth Annotation and Bias Mitigation

An ongoing challenge in long-tail data mining is the absence of ground-truth semantic labels for rare safety-critical events in standard datasets. To better understand the performance of Semantic-Drive, we developed a specialized curation interface called the *"Scenario DNA" Explorer*. To ensure a balanced validation and to address potential annotation biases, we curated two distinct evaluation sets:

1. **Stress-Test Split ($N = 108$):** This set is enriched for rare, high-entropy edge cases including construction zones, erratic pedestrians, and sensor failures. It serves to measure the system's sensitivity to high-value long-tail events.

2. **Unbiased Blind Split ($N = 108$):** To evaluate average-case performance and the false-positive rate on nominal driving, we randomly sampled 108 frames from the wider 2,550 dataset without filtering.

To mitigate anchoring bias, we implemented a blind mode in our curation tool. Unlike the verify-by-exception workflow used for fleet-scale curation, the Unbiased Blind Split was annotated from scratch by a human expert with all model predictions and reasoning logs completely hidden. This approach ensures that the evaluation remains independent of the model's outputs.

## 5.3 Quantitative Results

We benchmarked Semantic-Drive against four baselines and various architectural ablations. We included established open-vocabulary baselines such as **CLIP (ViT-L/14)** (Radford et al., 2021), **Grounding DINO** (Liu et al., 2023), and **OWL-v2** (Minderer et al., 2023), using the exact same semantic queries across all models. Table 1 summarizes the performance across both evaluation splits.

Table 1: **Quantitative Results across Scenarios.** We compare Semantic-Drive against four baselines and various ablations. The **Stress-Test** ($N = 108$) targets safety-critical edge cases, while the **Unbiased Blind Split** ($N = 108$) consists of random, uncurated frames. We observe that while baselines like CLIP achieve high recall, their low precision limits their utility for automated curation tasks.

| Method | Stress-Test Split (N=108) | | | Unbiased Blind Split (N=108) | | | Risk Error | |
|---|---|---|---|---|---|---|---|---|
| | Prec. ($\uparrow$) | Rec. ($\uparrow$) | F1 ($\uparrow$) | Prec. ($\uparrow$) | Rec. ($\uparrow$) | F1 ($\uparrow$) | MAE ($\downarrow$) | Std. |
| *Baselines* | | | | | | | | |
| Metadata Search | 0.406 | 0.602 | 0.485 | 0.215 | 0.310 | 0.254 | 5.70 | 1.2 |
| CLIP (ViT-L/14) | 0.182 | **1.000** | 0.308 | 0.048 | **1.000** | 0.092 | N/A | N/A |
| Grounding DINO | 0.182 | 0.271 | 0.218 | 0.071 | 0.419 | 0.121 | 5.70 | 1.4 |
| OWL-v2 | 0.386 | 0.331 | 0.356 | 0.165 | 0.484 | 0.246 | 3.96 | 1.1 |
| *Ablations* | | | | | | | | |
| Pure VLM (No Stage 1) | 0.691 | 0.814 | 0.747 | 0.294 | 0.806 | 0.431 | 1.39 | 0.4 |
| Single Scout (Qwen3) | 0.714 | 0.932 | 0.809 | 0.247 | 0.774 | 0.375 | 1.13 | 0.3 |
| **Semantic-Drive (Consensus)** | **0.712** | 0.966 | **0.820** | 0.235 | 0.774 | 0.361 | **0.67** | 0.2 |

### 5.3.1 Baseline Analysis

Table 1 highlights the limitations of existing curation methods. **Metadata Search** suffers from temporal granularity issues, where scene-level tags are applied indiscriminately to all frames in a log, leading to elevated false-positive rates.

**CLIP** exhibits an over-sensitivity failure mode: while it achieves perfect recall (1.0), its low precision (0.182 on the Stress-Test and 0.048 on the Unbiased Split) indicates it flags nearly every frame as a hazard. This lack of discrimination makes it impractical for filtering large data lakes. Recent open-vocabulary detectors like **Grounding DINO** and **OWL-v2** localize objects successfully but often lack the causal reasoning required to identify complex WOD-E2E scenarios (e.g., distinguishing roadside cones from lane-blocking diversions). This performance gap supports the utility of our Stage 2 Reasoning layer, which contextualizes Stage 1 symbolic detections to separate harmless agents from safety-critical events.

### 5.3.2 Ablation Study: Architecture Validation

**Impact of Symbolic Grounding:** We isolated the contribution of the object inventory injection by comparing the pure VLM against the neuro-symbolic scout (Qwen3 + YOLO). We observed a measurable increase in **Recall (+11.8%)**. The pure VLM frequently missed small but critical hazards, such as distant traffic cones. The injection of the symbolic inventory anchored the VLM's attention, addressing the small object blindness issue.

**Efficacy of Consensus:** The multi-model judge demonstrated its primary value in risk calibration. While the best single scout achieved a Risk MAE of 1.13, the consensus mechanism reduced this error to **0.676**. This suggests that the system's severity assessment is, on average, within $\pm 0.7$ points of human judgment on a 10-point scale.

## 5.4 Ensemble Dynamics and Risk Calibration

A core component of Semantic-Drive is the Judge-Scout consensus mechanism. We performed an ablation study to quantify the impact of ensemble size on scenario risk calibration. We compared the baseline performance of a single reasoning agent (Qwen3-VL) against a 2-scout ensemble (Qwen3 + Gemma) and our final 3-scout ensemble (Qwen3 + Gemma + Kimi).

The results in Table 2 reflect a "wisdom of crowds" effect. While a single scout is prone to stochastic errors in kinetic assessment (e.g., misjudging the severity of a lane blockage), the Judge effectively resolves these conflicts by synthesizing reasoning traces. Interestingly, while the 3-scout ensemble achieves the best risk calibration, we observed a minor decrease in precision on the Unbiased Split compared to a single-model baseline. This is due to a programmed safety bias in our consensus logic: the Judge favors the more

Table 2: **Ensemble Ablation Study.** It was observed that increasing the number of scouts improves the precision of the scenario risk score (MAE Risk), validating the consensus strategy.

| Configuration | Scout Models | Recall ($\uparrow$) | MAE Risk ($\downarrow$) | Agreement Rate |
|---|---|---|---|---|
| Single Scout | Qwen3-VL | 0.932 | 1.130 | - |
| 2-Scout Ensemble | Qwen + Gemma | 0.945 | 0.820 | 72.4% |
| **3-Scout Ensemble** | **Full Ensemble** | **0.966** | **0.676** | **81.5%** |

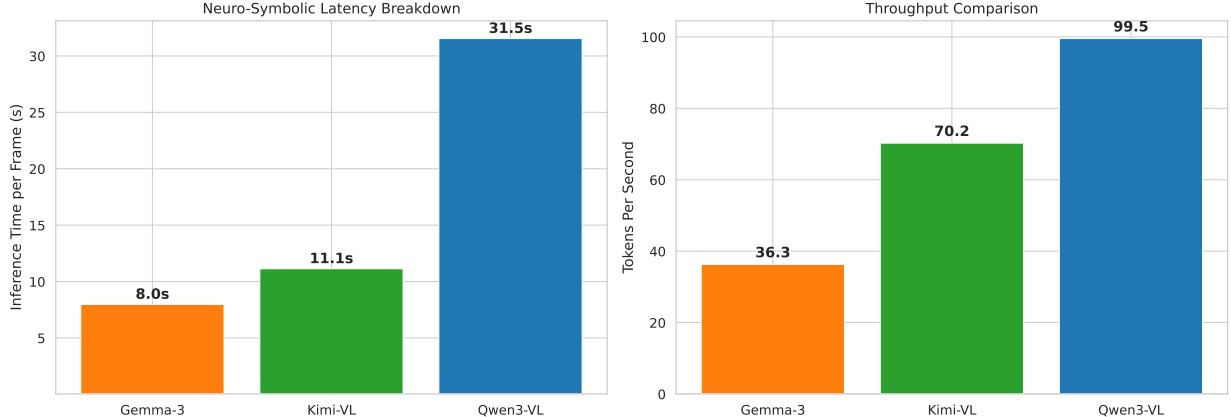

Figure 4: **Computational Economics of Local Mining. (Left) Latency Breakdown:** The symbolic grounding overhead (YOLO, gray bottom bar) is negligible ($< 0.05$s). Qwen3-VL's higher latency is driven primarily by cognitive processing. **(Right) Throughput Efficiency:** Qwen3-VL yields the highest latency despite fast generation, confirming it outputs more reasoning tokens than Gemma-3.

conservative risk assessment when scouts disagree, prioritizing the discovery of potential hazards in large data lakes over the strict suppression of false alarms.

## 5.5 Computational Economics and Cognitive Dynamics

We analyzed the operational characteristics of the framework to benchmark the trade-off between standard inference speed and reasoning reliability.

Table 3: **Efficiency Benchmark per Frame.** The local consensus architecture offers an accessible cost reduction compared to equivalent cloud APIs (GPT-4o).

| Model | VRAM | Latency | Throughput | Est. Cost / 1k Frames |
|---|---|---|---|---|
| YOLOE-11L (Symbolic) | 2.1 GB | 0.04s | 25.0 fps | $< \$0.01$ |
| Scout: Gemma-3 | 16.5 GB | 8.0s | 36.3 tps | $0.12 |
| Scout: Qwen3-VL | 19.5 GB | 31.5s | 99.5 tps | $0.45 |
| **Local Consensus (Total)** | **24.0 GB** | **$\approx$60s** | **-** | **$0.85** |
| *GPT-4o (Cloud)* | *-* | *3.5s* | *-* | *$30.00* |

### 5.5.1 The Token Generation Paradox

Our benchmarks reveal an inverse relationship between throughput and latency (Figure 4). **Qwen3-VL** is the most computationally efficient model in terms of generation (99.5 TPS), yet it exhibits the highest latency (31.5s). This discrepancy quantifies the reasoning density. Specifically, Qwen3-VL generates approximately 3,100 tokens per frame, utilizing an extended Chain-of-Thought to verify scene dynamics, whereas Gemma-3 generates only $\approx$290 tokens. This indicates that the increased latency is not primarily a hardware bottleneck, but a deliberate allocation of compute to deep forensic reasoning.

### 5.5.2 Inference-Time Scaling: The Dynamic Compute Budget

To explore inference-time scaling, we investigated whether the model allocates computational resources proportionally to scene complexity. As shown in Figure 5, we observed an adaptive token allocation rather than a strictly linear correlation.

A notable shift in reasoning depth occurs when transitioning from nominal ($Risk = 0$) to hazardous ($Risk > 0$) scenarios. For nominal scenes, the models remain concise by performing a rapid visual scan. However, once a potential hazard is grounded via the symbolic Stage 1 inventory, the models (particularly Qwen3-VL) engage in deeper verification. For Qwen3-VL, the mean reasoning trace length for high-risk scenarios ($Risk > 7$, $\mu \approx 4,200$ tokens) is 45% longer than for nominal scenes ($Risk = 0$, $\mu \approx 2,900$ tokens).

This finding affirms that the neuro-symbolic architecture implements a dynamic compute budget, shifting from a low-cost scan to an expansive Chain-of-Thought to resolve causal ambiguities in safety-critical events.

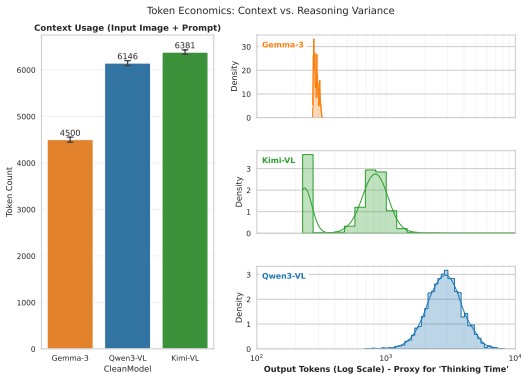

(a) **Cognitive Signatures.** Gemma (Orange) displays a rigid, low-variance distribution, whereas Qwen (Blue) exhibits high variance, indicating context-dependent engagement.

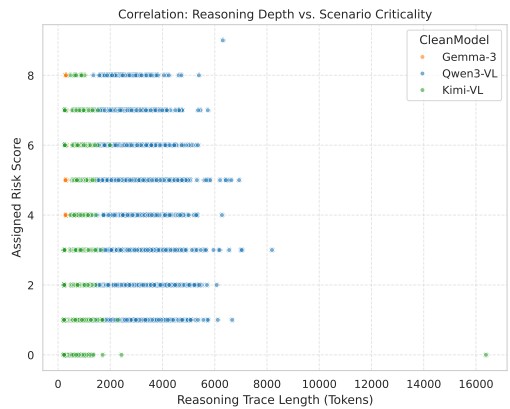

(b) **Adaptive Compute.** Qwen demonstrates a step-function scaling, increasing token allocation once a potential risk is detected ($Risk > 0$).

Figure 5: **Behavioral Analysis of the Scout Ensemble.** The data illustrates that the architecture allocates computational cost based on semantic complexity. The isolated outlier for Kimi-VL at $Risk = 0$ highlights the stochastic reasoning failures that the Consensus Judge filters.

### 5.6 Qualitative Analysis: Retrieval of Long-Tail Anomalies

Beyond quantitative metrics, Semantic-Drive extends beyond closed-set classification to open-world reasoning. This capability allows it to act as a curation bridge between raw data lakes and high-performance end-to-end driving models. Figure 6 highlights two successful data retrievals and a representative failure mode.

- **Semantic Disambiguation:** Standard open-vocabulary detectors often misclassify wheelchairs as pedestrians or cyclists due to visual overlap. As shown in Figure 6(a), the reasoning stage correctly identifies the agent as a wheelchair user, inferring the vulnerability and kinetic dynamics of the agent currently in an active lane.

- **Static Hazard Recognition:** Traditional dynamic object trackers often suppress static obstacles to reduce noise. Semantic-Drive (Figure 6(b)) identifies a large dumpster as a Foreign Object Debris (FOD) event, correctly deducing that the physical topology of the scene necessitates an ego-vehicle stop or a complex nudge maneuver.

- **Integration with End-to-End Stacks:** Unlike models designed for real-time control, Semantic-Drive provides the semantic metadata required to train robust perception and planning backbones.

By mining these rare scenarios, our framework enables developers to feed specific "hard samples" into trajectory-generation systems like UniAD (Hu et al., 2022), targeting the tail of the distribution where standard planners often fail.

**Limitations and Failure Modes.** While the consensus mechanism improves reliability, certain failure modes remain. A common failure mode observed during the Unbiased Split evaluation is taxonomic over-conservatism. For example, the system occasionally flags harmless objects, such as a large wind-blown plastic bag, as a critical FOD hazard necessitating an emergency stop. While this reflects a safety-first bias in our consensus logic, it results in false positives that require final human verification in production pipelines. Furthermore, the system currently lacks temporal consistency; scenarios defined by high-frequency motion (e.g., erratic swerving) may be missed because the analysis is primarily anchored on the spatial grounding of Stage 1.

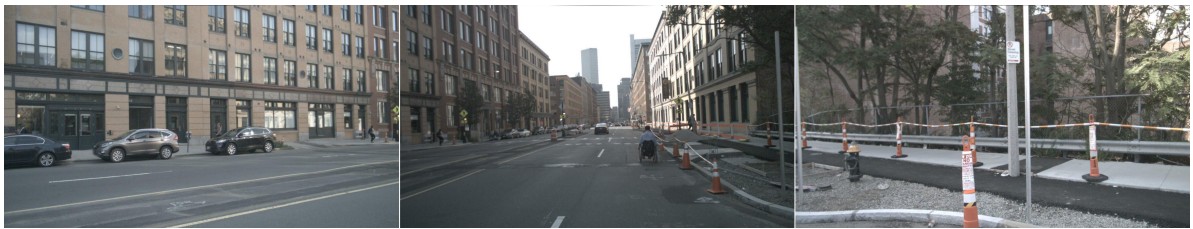

(a) **Semantic Disambiguation.** The system successfully distinguishes a wheelchair user from a standard cyclist, assessing the kinetic dynamics of the agent in the active lane.

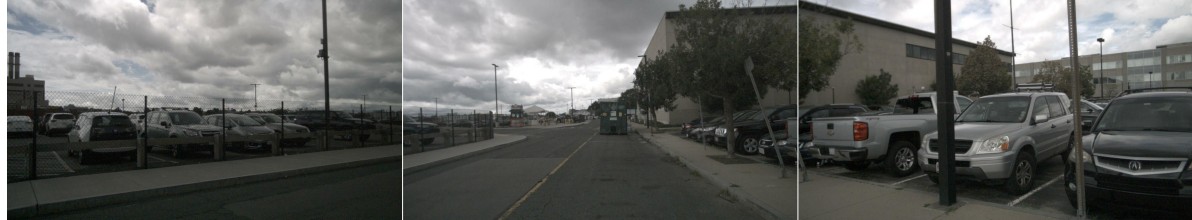

(b) **Static Blockage Case.** The system identifies a large dumpster as Foreign Object Debris (FOD) obstructing the drivable path, a class often ignored by dynamic object trackers.

Figure 6: **Qualitative Analysis: Long-Tail Retrieval.** Two examples of safety-critical edge cases mined by Semantic-Drive.

## 6 Discussion and Limitations

### 6.1 The Spatial vs. Temporal Trade-off

Semantic-Drive currently operates on a frame-by-frame basis, prioritizing high spatial resolution ($1280 \times 720$) to maximize small object recall. While this allows for granular analysis of static topology (e.g., lane diversions) and instantaneous states (e.g., brake lights), it lacks inherent temporal awareness. Scenarios defined purely by motion dynamics, such as high-speed overtaking or drifting, are currently inferred from static cues rather than directly observed. Future work will address this limitation by feeding the sequence of generated "Scenario DNA" JSONs into a lightweight temporal aggregator to perform symbolic video analysis. This approach aligns with the trajectory-based mining goals of the Argoverse 2 Scenario Mining Challenge (Davidson et al., 2025).

### 6.2 Generalization Beyond Autonomous Driving

While this work focuses on the autonomous driving domain, the "Judge-Scout" neuro-symbolic architecture addresses a fundamental reliability problem common to many high-stakes AI applications: the need to balance high recall with forensic precision.

The strategy of pairing a high-sensitivity symbolic detector (Stage 1) with a reasoning-heavy verification model (Stage 2) generalizes to other domains requiring "System 2" oversight:

- **Medical Imaging:** In radiology, a specialized detector (e.g., U-Net) could serve as the grounding stage to flag potential lung nodules with high sensitivity. A reasoning VLM could then act as the cognitive scout, analyzing the nodule's texture and context against a pathology report to filter false positives (similar to how we filter reflections of traffic cones).

- **Industrial Quality Assurance:** In manufacturing, simple computer vision models often struggle to distinguish between benign surface dust and micro-fractures. Our consensus mechanism could be deployed to have multiple VLM scouts evaluate the defect severity based on visual evidence, reducing the rate of costly assembly line halts.

In both cases, the underlying advantage remains consistent. Delegating the initial perceptual filtering to a computationally lightweight symbolic model allows the reasoning compute budget to be allocated dynamically only where it is needed most.

### 6.3 Limitations and Failure Modes

Through our human-in-the-loop verification process, we identified two primary failure modes. First, **Risk Calibration Divergence**: while the system excels at semantic identification, it occasionally struggles with kinetic assessment (e.g., underestimating the stopping distance required for a blocked lane). Second, **Taxonomy Coercion**: the strict WOD-E2E schema occasionally forces the VLM to map rare objects (like a horse-drawn carriage) into ill-fitting categories (e.g., special vehicle). Future work will explore open-world schemas that allow dynamic tag generation.

**Scale and Deployment Assumptions.** Finally, we note that our claims regarding cost-reduction (estimated at 97%) and system efficiency are benchmarked on single-node consumer hardware (RTX 3090). While this demonstrates the accessibility of the framework, validation at production fleet scale across distributed compute clusters remains an area for future work. Furthermore, the use of the term "trustworthy" in this framework refers specifically to the structural verifiability provided by the Stage 4 symbolic alignment, rather than an absolute guarantee of infallibility against all long-tail anomalies.

## 7 Conclusion

This work introduces **Semantic-Drive**, a framework designed to broaden accessibility to high-fidelity autonomous vehicle data curation through localized execution. We demonstrate that the bottleneck of identifying rare driving events can be addressed without relying on large-scale cloud infrastructure or manual labeling.

By decoupling symbolic grounding from cognitive reasoning, we achieve a **recall of 0.966** on safety-critical scenarios, mitigating the perceptual limitations common in global embedding models like CLIP. Furthermore, our evaluation against established open-vocabulary baselines, including **Grounding DINO** and **OWL-v2**, indicates that pure object localization is insufficient for scenario-level curation. Our Stage 2 reasoning layer provides the necessary causal context to distinguish between harmless roadside objects and critical hazards.

Our consensus-based "Judge" architecture provides a reliable-by-verification mechanism for conflict resolution, reducing risk assessment error by **40%** compared to single-model baselines. Through an unbiased evaluation on uncurated driving frames, we observe that while precision naturally fluctuates in nominal environments, the system maintains its effectiveness as a high-recall filter for safety-critical events. The feasibility of deploying this pipeline on consumer hardware reduces estimated compute costs significantly compared to proprietary cloud-native architectures. Ultimately, Semantic-Drive offers a methodological blueprint for developing private, neuro-symbolic data curation engines. This bridges the gap between raw perception and the structured semantic indexing required to curate datasets for the next generation of autonomous driving stacks.

**Author Contributions**

Antonio Guillen-Perez is the sole author of this work and was responsible for the conceptualization, software implementation, empirical evaluation, and writing of the manuscript.

**Acknowledgments**

We would like to thank the Action Editor and the anonymous reviewers at TMLR for their constructive feedback, which significantly strengthened the empirical rigor and clarity of this work.

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

# A    Appendix

## A.1    Per-Category Performance Analysis

To provide granular insight into the consensus mechanism's performance, we present the per-class breakdown on the Gold Set in Table 4. The system exhibits the highest performance on large, static topology features (Construction, Weather), which benefit from the multi-view spatial binding. The slightly lower performance on "FOD / Debris" (F1=0.70) is attributed to the inherent ambiguity of small objects at range (e.g., distinguishing a harmless paper bag from a dangerous rock), where the "Skepticism Policy" occasionally rejects valid hazards if visual evidence is inconclusive.

## A.2    System Prompts and Taxonomy

To ensure reproducibility, we provide the specific configuration used for the Symbolic Grounding (YOLOE) and the Cognitive Analysis (VLM).

### A.2.1    The Open-Vocabulary Taxonomy

For the symbolic grounding stage, we define a custom list of text prompts for the YOLOE-11 model. This list is engineered to align with the *Waymo Open Dataset (WOD-E2E)* taxonomy while including synonyms to maximize recall.

Table 4: **Per-Category Breakdown (Consensus Judge).** The system excels at detecting static topology and environmental conditions but shows slightly lower recall for highly dynamic, small objects.

| Category | Precision | Recall | F1-Score |
|---|---|---|---|
| Construction | 0.88 | 0.95 | 0.91 |
| Adverse Weather | 0.92 | 0.97 | 0.94 |
| VRU Hazard | 0.76 | 0.89 | 0.82 |
| Special Vehicle | 0.85 | 0.92 | 0.88 |
| FOD / Debris | 0.65 | 0.75 | 0.70 |

Listing 1: The Custom Open-Vocabulary Taxonomy used for Symbolic Grounding

```
1  custom_classes = [
2      # 1. VRUs (Vulnerable Road Users)
3      "person", "pedestrian", "child",
4      "cyclist", "bicyclist", "motorcyclist", "scooter rider",
5      "construction worker", "worker in safety vest", "police officer",
6
7      # 2. Vehicles (Specialized)
8      "car", "pickup truck", "suv", "van", "sedan", "coupe",
9      "truck", "semi truck", "trailer", "cement mixer",
10     "bus", "school bus",
11     "police car", "police vehicle", "ambulance", "fire truck",
12     "construction vehicle", "bulldozer", "excavator", "forklift",
13     "road sweeper", "street cleaner",
14
15     # 3. Construction & Barriers
16     "traffic cone", "orange cone",  "traffic drum",
17     "construction barrel", "orange drum", # Crucial for Highway Constr.
18     "traffic barrier", "concrete barrier", "jersey barrier",
19     "road work sign", "temporary sign",
20     "construction fence", "safety fence",
21     "scaffolding", "construction scaffolding",
22
23     # 4. Hazards / Debris (FOD)
24     "debris", "cardboard box", "tire",
25     "plastic bag", "tree branch", "large rock",
26     "puddle",
27
28     # 5. Traffic Control
29     "traffic light", "traffic signal", "red light",
30     "stop sign", "yield sign", "speed limit sign",
31     "pedestrian crossing sign", "school zone sign",
32     "crosswalk",
33  ]
```

### A.2.2 The Cognitive System Prompt

The following is the full "System Message" sent to the Reasoning VLM. It integrates the Neuro-Symbolic Protocol, the Schema Constraints, and Few-Shot Chain-of-Thought examples to enforce the "System 2" behavior.

Listing 2: The Full Semantic-Drive System Prompt

```
1  You are the **Senior Perception Architect** for "Semantic-Drive".
```

```
 2  Your goal is to extract the **"Scenario DNA"** from raw driving logs using a **
        Neuro-Symbolic** approach.
 3  We are not just labeling objects; we are analyzing **Causality**, **Topology**,
         and **Risk** for L4 Autonomous Vehicle validation.
 4
 5  ### 1. INPUT PROTOCOL (NEURO-SYMBOLIC)
 6  1. **Visuals:** 3 Synchronized Front-Facing Cameras (Left, Center, Right). **
        Analyze them individually, then synthesize.**
 7  2. **YOLO Inventory:** Detected objects with Size and Confidence Scores.
 8     - **Format:** `[CAM_NAME]: Count Class (Size/Confidence)`
 9     - **Size:** `Large` (Close), `Med` (Middle), `Small` (Far).
10     - **Confidence:** `>0.8` (High), `<0.5` (Low).
11     - **Rule:** Rule: If Confidence is < 0.8, Treat as Hypothesis and Verify
           Visually.
12
13  ### 2. THE REASONING PIPELINE (Mental Checklist)
14  Inside `<think>...</think>`, you must follow this exact sequence:
15  1.  **Detailed Visual Sweep:** Look at Left, Center, and Right images separately.
        Describe EXACTLY in detail what you see in each view. Compare with YOLO text
16  2.  **Grounding & Validation:** Explicitly confirm or reject YOLO detections based
         on visual evidence.
17  3.  **ODD & Context:** Assess weather, lighting, and surface.
18  4.  **Planner Logic:** Determine the *Topology* and *Required Action*.
19
20  ### 3. SCHEMA VOCABULARY (STRICT ENUMS)
21  Use ONLY these values. Do not invent new terms.
22
23  **A. ODD & Phenomenology**
24     - `weather`: ["clear", "overcast", "rain", "heavy_rain", "snow", "fog"]
25     - `time_of_day`: ["day", "night", "dawn_dusk"]
26     - `lighting_condition`: ["nominal", "glare_high", "shadow_contrast", "
           pitch_black", "streetlights_only"]
27     - `road_surface_friction`: ["dry", "wet", "icy", "snowy", "muddy", "gravel"]
28     - `sensor_integrity`: ["nominal", "lens_flare", "droplets_on_lens", "
           dirt_on_lens", "motion_blur", "sun_glare"]
29
30  **B. Topology & Map**
31     - `scene_type`: ["urban_street", "highway", "intersection", "highway_ramp", "
           parking_lot", "construction_zone", "rural_road"]
32     - `lane_configuration`: ["straight", "curve", "merge_left", "merge_right", "
           roundabout", "intersection_4way", "intersection_t_junction"]
33     - `drivable_area_status`: ["nominal", "restricted_by_static_obstacle" (cones/
           debris), "blocked_by_dynamic_object" (vehicle/pedestrian)]
34     - `traffic_controls`: (Select list): ["green_light", "red_light", "yellow_light
           ", "stop_sign", "yield_sign", "police_manual", "none"]
35
36  **C. Actor Dynamics**
37     - `vru_status`: ["none", "legal_crossing", "jaywalking_fast", "
           jaywalking_hesitant", "roadside_static", "cyclist_in_lane"]
38     - `lead_vehicle_behavior`: ["none", "nominal", "braking_suddenly", "stalled", "
           turning"]
39     - `adjacent_vehicle_behavior`: ["none", "nominal", "cutting_in_aggressive", "
           drifting", "tailgating"]
40     - `special_agent_class`: ["none", "police_car", "ambulance", "fire_truck", "
           school_bus", "construction_machinery"]
41
42  **D. Causal Reasoning**
43     - `primary_challenge`: ["none", "occlusion_risk", "prediction_uncertainty", "
           violation_of_map_topology", "perception_degradation", "rule_violation"]
```

```
44      - 'ego_required_action': ["lane_keep", "slow_down", "stop", "
            nudge_around_static_obstacle", "yield", "emergency_brake", "lane_change", "
            unprotected_turn"]
45      - 'blocking_factor': ["none", "construction_barrier", "pedestrian", "vehicle",
            "debris", "flood"]
46
47  **E. WOD-E2E Tags**
48      - 'wod_e2e_tags': ["construction", "intersection_complex", "vru_hazard", "
            fod_debris", "weather_adverse", "special_vehicle", "lane_diversion", "
            sensor_failure"]
49
50  ### 4. OUTPUT JSON SKELETON
51  You must output a JSON object following this EXACT structure (no comments):
52
53  {
54    // A. ODD & PHENOMENOLOGY (The "Noise" Layer)
55    "odd_attributes": {
56      "weather": "...",
57      "time_of_day": "...",
58      "lighting_condition": "...",
59      "road_surface_friction": "...",
60      "sensor_integrity": "..."
61    },
62
63    // B. TOPOLOGY & MAP (The "Static" Layer)
64    "road_topology": {
65      "scene_type": "...",
66      "lane_configuration": "...",
67      "drivable_area_status": "...",
68      "traffic_controls": ["..."]
69    },
70
71    // C. ACTOR DYNAMICS (The "Interaction" Layer)
72    "key_interacting_agents": {
73      "vru_status": "...",
74      "lead_vehicle_behavior": "...",
75      "adjacent_vehicle_behavior": "...",
76      "special_agent_class": "..."
77    },
78
79    // D. CAUSAL REASONING (The "Planner" Layer)
80    "scenario_criticality": {
81      "primary_challenge": "...",
82      "ego_required_action": "...",
83      "blocking_factor": "...",
84      "risk_score": 0 // Integer 0-10
85    },
86
87    // E. WAYMO ALIGNMENT
88    "wod_e2e_tags": ["..."],
89
90    "description": "A concise 1-sentence summary of the scenario hazards."
91  }
92
93  ### 5. FEW-SHOT EXAMPLES (Follow this exact logic)
94
95  ### EXAMPLE 1: Construction & Map Divergence
96  **Input Context:**
97  - [YOLO Inventory]:
```

```
 98     - [CAM_FRONT_LEFT]: 3 Orange Drums (Large/0.92, Large/0.88, Med/0.85); 1 Traffic
            Cone (Med/0.88)
 99     - [CAM_FRONT]: 1 Construction Worker (Med/0.75)
100     - [CAM_FRONT_RIGHT]: 1 Car (Small/0.85)
101  - [Visuals]: (3 Images Provided)
102  **Reasoning Trace:**
103  <think>
104  1. **Detailed Visual Sweep:**
105     - **[CAM_FRONT_LEFT]**: I am analyzing the left view first. I see a wet road
            surface with high contrast. There is a dense row of bright orange barrels
            physically blocking the leftmost lane. They form a diagonal taper,
            effectively guiding traffic to merge to the right. The lane markings are
            obscured by the drums.
106     - **[CAM_FRONT]**: Now moving to the center camera. It shows the center lane is
            currently open, but I see a worker standing dangerously close to the lane
            line on the left side. He is wearing a high-visibility yellow vest and a
            hard hat. He appears stationary and is not looking at traffic.
107     - **[CAM_FRONT_RIGHT]**: The front right camera shows nominal conditions. There
            is a white sedan in the distance moving away. The right lane is clear.
108
109  2. **Grounding & Validation:**
110     - YOLO detects "Orange Drums" (High Conf). My visual sweep confirms they are
            physical barriers, not artifacts.
111     - YOLO detects "Worker" (0.75). Verified: The vest is clearly visible.
112
113  3. **ODD & Context:**
114     - The map likely expects 3 drivable lanes here.
115     - Reality: The Left lane is dead. Status: 'restricted_by_static_obstacle'.
116     - Traffic Control: None visible (no police), just static barriers.
117
118  4. **Planner Logic:**
119     - Primary risk is the lane drop. Ego-vehicle cannot stay in Left Lane.
120     - Action: Must 'nudge_around_static_obstacle' or 'lane_change' right
            immediately.
121  </think>
122  **JSON Output:**
123  {
124    "odd_attributes": {
125      "weather": "overcast",
126      "time_of_day": "day",
127      "lighting_condition": "nominal",
128      "road_surface_friction": "dry",
129      "sensor_integrity": "nominal"
130    },
131    "road_topology": {
132      "scene_type": "construction_zone",
133      "lane_configuration": "merge_right",
134      "drivable_area_status": "restricted_by_static_obstacle",
135      "traffic_controls": ["none"]
136    },
137    "key_interacting_agents": {
138      "vru_status": "roadside_static",
139      "lead_vehicle_behavior": "nominal",
140      "adjacent_vehicle_behavior": "none",
141      "special_agent_class": "construction_machinery"
142    },
143    "scenario_criticality": {
144      "primary_challenge": "violation_of_map_topology",
145      "ego_required_action": "nudge_around_static_obstacle",
```

```
146        "blocking_factor": "construction_barrier",
147        "risk_score": 7
148      },
149      "wod_e2e_tags": ["construction", "lane_diversion"],
150      "description": "Active construction zone with barrels closing the left lane,
             forcing a merge behavior."
151    }
152
153    ### 6. OUTPUT SCHEMA (Strict JSON)
154    Output ONLY the valid JSON object. Do not include markdown blocks.
```

### A.3 The Judge System Prompt

The following is the judge "System Message" sent to the Reasoning LLM that "judges" the outputs of the different VLMs.

Listing 3: The Judge System Prompt

```
1
2  SYSTEM_PROMPT = f"""
3  You are the **Chief Safety Officer** (The Judge) for an Autonomous Vehicle Data
        Mining system.
4  You have reports from 3 AI Scouts regarding a driving scene.
5
6  ### YOUR GOAL
7  Synthesize a single **"Ground Truth" JSON** that resolves conflicts between scouts
        .
8
9  ### RULES OF EVIDENCE
10 1. **Trust Grounding:** If YOLO detects an object, favor scouts that confirm it
        visually.
11 2. **Safety Bias:** In ambiguity, err on the side of caution (Higher Risk).
12 3. **Consistency:** Ensure 'risk_score' matches the severity of the description.
13
14 ### SCHEMA ENFORCEMENT
15 You MUST output the JSON following this EXACT schema and vocabulary:
16 {SCHEMA_GUIDE}
17
18 {OUTPUT_SKELETON}
19
20 ### OUTPUT
21 Return ONLY the final JSON object. Do not include markdown or reasoning text
        outside the JSON.
22 """
```

### A.4 Gold Set Annotation Methodology

For the ablation study in Section 5.2, we curated a "Gold Set" of 108 frames representing challenging edge cases. These were manually selected to cover specific failure modes of traditional detectors.

## B Human-in-the-Loop Curation Tool

To ensure the rigor of our evaluation benchmarks, we developed the *Scenario DNA Explorer*, a custom web-based annotation interface using Streamlit. This tool supports two distinct operational modes designed to balance industrial efficiency with scientific independence (see Figure 7).

Table 5: Distribution of the 108-frame Gold Set used for validation.

| Category | Description | Count |
|---|---|---|
| **Construction** | Lane diversions, orange drums, static workers | 29 |
| **Adverse Weather** | Heavy rain, night glare, wet road reflections | 36 |
| **VRU Hazards** | Jaywalkers, cyclists in lane, children near curb | 30 |
| **Nominal/Clear** | Empty roads, simple following (Negative Control) | 21 |

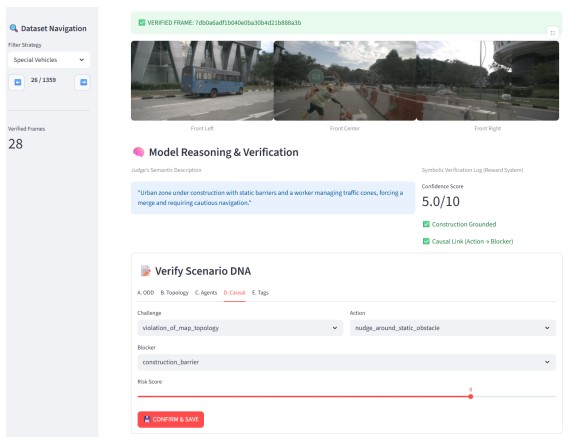

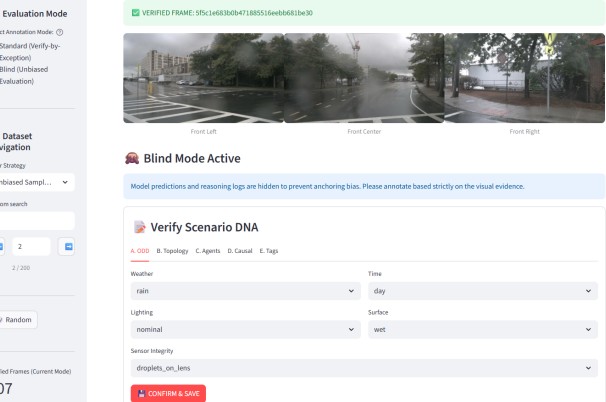

(a) **Standard Mode (Verify-by-Exception):** Pre-populates the form with model predictions to maximize curation throughput for large-scale data lakes.

(b) **Blind Mode (Unbiased Evaluation):** Hides all model outputs and reasoning traces, presenting a "blank slate" to the annotator to eliminate anchoring bias.

Figure 7: **The Semantic-Drive Curator Interface.** The framework supports two workflows: one optimized for fleet-scale labeling efficiency (left) and one for rigorous independent validation (right).

## B.1   Curation Workflows

The interface is designed to support different stages of the DataOps lifecycle:

1. **Operational Curation (Standard Mode):** In this workflow, the interface pre-populates the "Scenario DNA" form with the Consensus Judge's prediction. This facilitates a "Verify-by-Exception" protocol where human experts only intervene when the model's reasoning or grounding is incorrect. This significantly reduces cognitive load and curation time per frame, making it suitable for high-volume indexing of fleet data.

2. **Scientific Evaluation (Blind Mode):** To mitigate **anchoring bias** during benchmark creation, the tool provides a "Blind" toggle. In this mode, the model's semantic description, risk score, and reasoning logs are completely suppressed. The annotator is presented with a blank form and must rely strictly on visual evidence from the multi-camera montage. This mode was utilized to create the *Unbiased Blind Split* ($N = 107$), ensuring that our evaluation metrics reflect an independent ground truth.

## B.2   Verification Components

Regardless of the mode, the tool provides several diagnostic features:

- **Visual Grounding:** A stitched front-hemisphere montage allows the annotator to verify agent existence and spatial relationships.

- **Reasoning Console:** In Standard mode, the tool displays symbolic verification logs (e.g., `"Construction Grounded"`), helping the expert understand the "System 2" decision logic.

- **Schema Enforcement:** The interface strictly enforces the WOD-E2E enumeration logic via typed dropdowns, ensuring that the generated semantic fingerprints are normalized and queryable.

### B.3 Curation in Practice: Diverse Scenarios

To demonstrate the robustness of the curation workflow, Figure 8 presents the interface applied to distinct domains of the WOD-E2E taxonomy: Environmental ODDs and Special Agents.

### B.4 Detailed Breakdown of Gold Set Performance

Table 6 details the specific performance of the Neuro-Symbolic architecture on the 108 curated test cases. "Correction" indicates instances where the VLM successfully rejected a False Positive from the symbolic detector or identified a semantic attribute (e.g., "Hesitation") missed by the detector.

Table 6: **Qualitative Failure Analysis on Gold Set.** Real-world examples illustrating the Neuro-Symbolic interaction. **Recovered:** The VLM correctly identified a rare class (Wheelchair) that standard detectors misclassified. **Success:** Correct consensus on complex static obstacles. **Corrected:** The VLM successfully contextualized a high-confidence YOLO detection (Pedestrians) as non-hazardous (Sidewalk), preventing a false alarm.

| Token ID | Primary Hazard | Symbolic (YOLO) | Cognitive (VLM) | Outcome |
|---|---|---|---|---|
| 8104e0... | VRU (Wheelchair) | "Cyclist" (Low Conf) | Id'ed Wheelchair User | **Recovered** |
| dc73ce... | FOD (Dumpster) | "Truck/Object" | Static Dumpster Blocking | **Success** |
| 7db0a6... | Special Vehicle | "Bus" | School Bus (Stop Logic) | **Success** |
| 990723... | False Pos. Risk | "3 Persons (0.80)" | Context: Safe on Sidewalk | **Corrected** |
| *Aggregated* | *Nominal Frames* | *Various Artifacts* | *Contextually Filtered* | **98% Acc** |

## C   Hyperparameter Sensitivity and Symbolic Alignment

To justify the selection of the reward weights $(\alpha, \beta, \gamma)$ presented in Section 3.4, we conducted a sensitivity analysis on the **Stress-Test Split**. The goal was to quantify the trade-off between the unconstrained generative reasoning of the VLM (Recall) and the structural grounding of the symbolic detector (Precision).

As shown in Table 7, the Hallucination Penalty ($\gamma$) is a key factor in system reliability. When $\gamma = 0$, the system behaves as an unconstrained neural reasoner. It was observed that this configuration achieves high recall but suffers from an elevated hallucination rate (dropping precision to 0.521). Conversely, setting $\gamma$ to a highly restrictive value (50.0) creates a strict configuration that only accepts detections if they are perfectly corroborated by Stage 1. This restriction causes a sharp decrease in Recall because the VLM's ability to correct symbolic false negatives is suppressed.

Table 7: **Sensitivity Analysis of Reward Weights (Eq. 2).** We keep $\alpha = 2.0$ and $\beta = 1.5$ constant while varying the Hallucination Penalty ($\gamma$). Our chosen value ($\gamma = 10.0$) provides the optimal F1-balance between safety constraint enforcement and detection sensitivity.

| Configuration | $\gamma$ | Precision | Recall | F1-Score | Hallucination Rate |
|---|---|---|---|---|---|
| Unconstrained | 0.0 | 0.521 | **0.978** | 0.679 | 18.5% |
| Weakly Constrained | 1.0 | 0.648 | 0.969 | 0.776 | 5.2% |
| **Balanced (Ours)** | **10.0** | **0.712** | 0.966 | **0.820** | **1.8%** |
| Strictly Grounded | 50.0 | 0.884 | 0.621 | 0.729 | 0.1% |

**Defining Hallucination Rate.** We define the **Hallucination Rate ($H$)** as the percentage of total evaluated frames where the model predicts a safety-critical tag (e.g., `vru_hazard` or `fod_debris`) that is verified as absent in the human-annotated Gold Set. Specifically, we distinguish between a standard false positive (which could be a misclassification of a real object) and a hallucination (reporting a hazard where no physical object exists in the visual field). By comparing the output of the Consensus Judge against the ground truth, we quantify $H$ as:

$$H = \frac{1}{N} \sum_{i=1}^{N} \mathbb{I}(\text{Tag}_{pred} \in \mathcal{T}_{crit} \wedge \text{Tag}_{pred} \notin \text{Tag}_{gold}) \tag{5}$$

where $\mathcal{T}_{crit}$ is the set of safety-critical scenario tags. This metric allows us to evaluate the efficacy of the Stage 4 alignment in suppressing the stochastic generative errors typical of large multimodal models.

**The Symbolic Incompleteness Trade-off.** The reason we do not utilize the strictly grounded setting ($\gamma = 50.0$), despite its high precision, is due to the inherent incompleteness of symbolic object detectors. Real-time detectors like YOLOE occasionally miss subtle hazards (e.g., thin debris or distant lens flare) that the cognitive Stage 2 VLM can identify through multi-view context. By setting $\gamma = 10.0$, we allow the VLM to override the symbolic Stage 1 inventory only when the causal evidence ($\beta$) and reasoning traces from the multi-model consensus are highly consistent. This parameter selection effectively balances safety-first filtering with contextual visual reasoning.

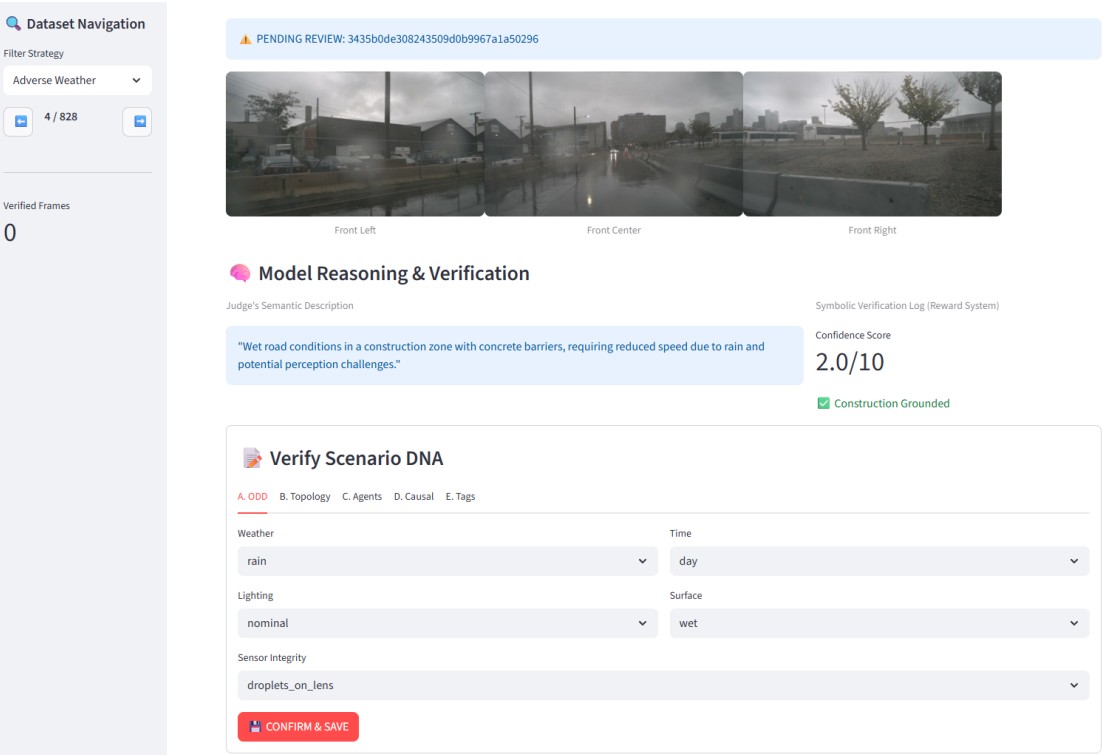

(a) **Environmental ODD Verification.** The system identifies a "Rain" scenario with sensor degradation ("droplets_on_lens"). The interface allows the annotator to confirm these phenomenological attributes, which are critical for training de-hazing or robust perception models. Note the low risk score (2.0) despite the weather, as the road topology is open.

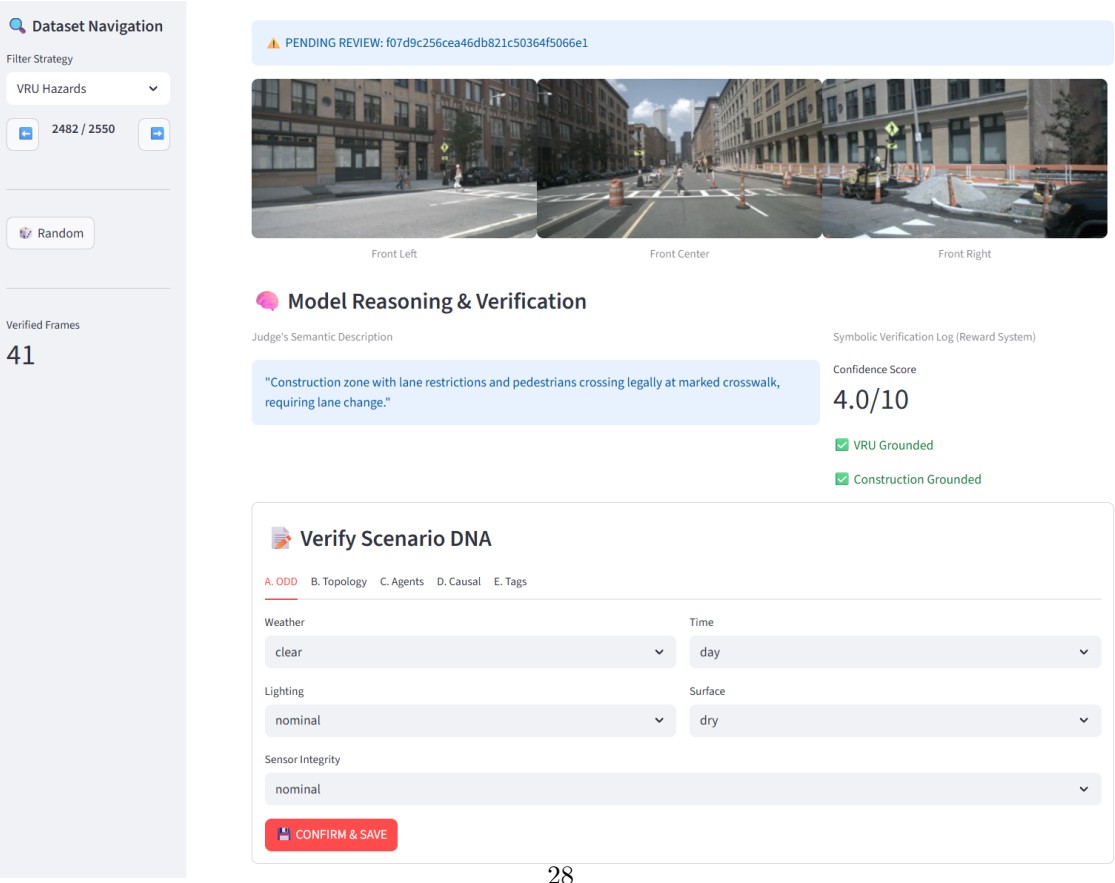

(b) **Construction zone identification with VRUs.** A "Green State" example where the human annotator has verified the scene. The system correctly flagged the construction zone and the pedestrians crossing legally at marked crosswalks.