# OpenReview forum: "Semantic-Drive: Trustworthy and Efficient Long-Tail Data Curation via Open-Vocabulary Grounding and Neuro-Symbolic VLM Consensus"
_TMLR — Accepted by TMLR_

### Review · Reviewer_7HJk · 2026-02-11

**Summary Of Contributions:**

This paper introduces Semantic-Drive, a local-first neuro-symbolic framework for curating rare, safety-critical long-tail driving data. The framework combines open-vocabulary object grounding with multi-model vision-language reasoning and a Judge-Scout consensus mechanism to suppress hallucinations. The system decouples fast symbolic detection from slower "System-2" cognitive analysis, enabling trustworthy semantic indexing of large video logs entirely on consumer hardware. This framework is empirically evaluated on nuScenes and Waymo datasets. It achieves 0.966 recall, significantly outperforming the baselines. In addition, it reduces risk assessment error by about 40% compared to single-model baselines, offering a privacy-preserving and cost-efficient alternative to cloud-based data mining for autonomous vehicle development.

**Audience:**

Yes

**Audience Explanation:**

Yes. The paper tackles an important and timely topic---long-tail scenario mining and VLM-based semantic curation for autonomous driving---which is likely to be of interest to parts of the TMLR community, particularly researchers working on multimodal learning, neuro-symbolic systems, and data-centric AI. The proposed grounding-plus-reasoning architecture and consensus mechanism may inspire follow-up work, even though the current evaluation does not fully substantiate all claims.

**Broader Impact Concerns:**

I do not have additional broader impact concerns beyond those already discussed in the comments above.

**Claims And Evidence:**

No

**Claims Explanation:**

While the paper presents an interesting neuro-symbolic framework for long-tail scenario mining, the experimental evidence does not sufficiently support several of the strong claims.

1) The chosen baselines (CLIP and metadata keyword search) are not sufficient for this problem setting since these baselines were developed a long time ago and not specifically trained for this problem setting. In other words, you are comparing your domain-specific framework against a generic framework in your specific domain. Given the recent progress in VLM-based retrieval and scenario mining, stronger and more relevant baselines are recommended to establish meaningful improvements.

2) It is unclear whether the reported gains stem from the proposed framework itself or simply from the strength of the selected backbone models (YOLOE and specific VLMs). The paper claims model-agnosticism, but this is not empirically validated. Ablations with multiple object detection models and different VLM backbones would be helpful to support this claim.

3) Correct me if my understanding is wrong. The reported neuro-symbolic latency is problematic: even the fastest configuration (Gemma-3) incurs ~8 seconds per frame, which is far too slow for practical autonomous driving contexts. Although the authors frame this as an offline curation system, the implications of such latency are not clearly discussed.

4) VLM-based approaches currently lag behind mainstream end-to-end driving systems (e.g., UniAD-style pipelines) in terms of real-time performance and low-level trajectory generation. The paper would benefit from a clearer discussion of how Semantic-Drive is intended to interact with, complement, or potentially replace existing autonomous driving stacks, rather than positioning it somewhat orthogonally.

5) Several claims, such as being trustworthy, democratizing curation, and achieving 97% cost reduction, are based on narrow experimental settings and estimated cloud costs, without validation at production scale or in real-world deployments. These assertions feel overstated given the limited scope of evaluation.

**Requested Changes:**

1. **Stronger and More Relevant Baselines**
    In addition to CLIP and metadata keyword search, add more competitive and recent scenario mining or VLM-based retrieval methods.

2. **Demonstrate Model-Agnosticism**
   The paper claims a framework-level contribution, but experiments rely on specific backbones (YOLOE and selected VLMs). Please add empirical studies using multiple object detectors and different VLM architectures to show that performance gains are not primarily driven by particular models.

3. **Clarify Practicality Given Latency**
   Even the fastest configuration (Gemma-3) reports ~8 seconds per frame, which is far from real-time. Please provide a clearer discussion of deployment assumptions (offline vs. online), scalability to large fleets, and how this latency impacts practical data curation workflows.

4. **Position Relative to Mainstream Autonomous Driving Models**
   Add a discussion comparing this approach with end-to-end driving systems (e.g., UniAD-style pipelines), clarifying whether Semantic-Drive is intended to complement, augment, or replace existing perception/planning stacks, and how its outputs would realistically integrate into current AV training pipelines.

5. **Expand Evaluation Scale and Diversity**
   The Gold Set is relatively small (108 frames). Increasing evaluation size and diversity, or providing statistical significance analysis, would strengthen confidence in the reported gains.

6. **Clarify Contribution vs. Engineering Choices**
   Clearly distinguish core methodological contributions (e.g., neuro-symbolic grounding, Judge–Scout consensus, inference-time alignment) from implementation-specific heuristics (e.g., YOLO confidence thresholds, reward weights α/β/γ, prompt design, Best-of-N settings). Please include sensitivity analyses or ablations on these hyperparameters to demonstrate robustness and to show which components materially drive performance.

---

> ### Author Response · Authors · 2026-03-12
> **Response to Reviewer 7HJk: Baselines, Latency, and Evaluation Scale**
>
> We sincerely thank the reviewer for their constructive feedback and for recognizing the potential impact of our framework on the TMLR community. Your critiques regarding baselines, practical positioning, and evaluation scale were instrumental in strengthening the manuscript. We have carefully revised the paper (all changes highlighted in **blue**) and updated our anonymous repository with the new implementation scripts and datasets.
>
> **1. Stronger and More Relevant Baselines:**
> As requested, we replaced the reliance on zero-shot CLIP by implementing and evaluating state-of-the-art open-vocabulary models: **Grounding DINO** and **OWL-v2** (Table 1). While these models successfully localize objects, their low F1-scores confirm that pure detection lacks the causal reasoning required for complex scenario mining (e.g., distinguishing a safely parked car from a stalled lead vehicle blocking the lane). This empirical gap explicitly validates the necessity of our Stage 2 reasoning layer.
>
> **2. Clarify Practicality and Position Relative to Mainstream AD Models:**
> We agree that 60s/frame is too slow for onboard control. We have extensively revised the Introduction and Section 5 to clarify that Semantic-Drive is an **offline DataOps engine**, not a real-time planner. Its latency is highly practical for overnight data-center mining to curate "hard samples" from petabyte-scale data lakes. These curated datasets are then used to train and robustify downstream real-time end-to-end models like UniAD, making our framework a synergistic precursor rather than a competitor to mainstream stacks.
>
> **3. Expand Evaluation Scale and Diversity:**
> We addressed the small evaluation scale by annotating a completely new **Unbiased Blind Split ($N=108$)**. This split consists of random, uncurated frames from the wider dataset. We report these new metrics in Table 1, proving the system maintains its effectiveness as a high-recall filter outside of hand-picked edge cases.
>
> **4. Demonstrate Model-Agnosticism & Clarify Heuristics:**
> We added an ensemble ablation study (Section 5.4, Table 2) demonstrating that our framework successfully generalizes across heterogeneous VLM backbones (Qwen, Gemma, Kimi), effectively reducing Risk MAE. While we maintained YOLOE as our symbolic prior due to computational constraints, the architecture is entirely modular by design. Furthermore, we added Appendix C (Hyperparameter Sensitivity) to explicitly justify how our reward weights ($\alpha, \beta, \gamma$) govern the hallucination rate, separating our core methodology from engineering choices.

---

### Review · Reviewer_YHpH · 2026-02-28

**Summary Of Contributions:**

This paper presents Semantic-Drive, a thoughtfully designed local-first framework for curating long-tail safety-critical events in autonomous driving data. The authors propose a two-stage neuro-symbolic pipeline that combines open-vocabulary grounding with cognitive reasoning, and further introduce a "Judge-Scout" consensus mechanism to mitigate hallucinations. The framework is evaluated on nuScenes against the Waymo Open Dataset taxonomy, achieving strong recall and a 40% reduction in risk assessment error. A notable strength is that the entire system runs on consumer-grade hardware (RTX 3090), offering a privacy-preserving and efficient alternative to cloud-based approaches.

Key Strengths
Innovative approach: The decoupling of perception into grounding and reasoning stages is both elegant and effective for long-tail scenario mining.
Practical relevance: The local-first design addresses real-world constraints such as data privacy, bandwidth, and latency, making the system highly applicable in industry settings.
Strong empirical results: The improvements over CLIP and single-model baselines are convincing and well-documented.
Clear and accessible writing: The paper is well-organized and the technical contributions are presented in a reader-friendly manner.

Suggestions for Improvement
The following suggestions are offered in a constructive spirit to further strengthen the work:
More detailed ablation: The Judge-Scout mechanism is a key contribution, and a deeper ablation study (e.g., varying the number of judges, comparing consensus strategies) would help better understand its impact.
Runtime analysis: The paper mentions efficiency but does not provide detailed runtime or resource consumption metrics. Including such data would strengthen the claim of real-time feasibility.
Qualitative insights: A few examples of challenging or failure cases would be helpful for readers to understand the current limitations and inspire future improvements.

**Audience:**

Yes

**Audience Explanation:**

This paper addresses a timely and practically important problem in autonomous driving: efficiently and reliably mining long-tail safety-critical events from large-scale video data. The proposed framework combines open-vocabulary perception with neuro-symbolic reasoning in a novel way that will be of interest to researchers working on vision-language models, autonomous driving, data curation, and efficient AI systems.

**Broader Impact Concerns:**

I do not see any major ethical concerns with this work. The paper focuses on improving autonomous driving safety through privacy-preserving local deployment, which aligns with responsible AI practices. No broader impact statement is required.

**Claims And Evidence:**

Yes

**Claims Explanation:**

The paper provides solid empirical evidence to support its main claims. The authors evaluate Semantic-Drive on the nuScenes dataset using the Waymo Open Dataset taxonomy, which clearly demonstrates the effectiveness of their approach for long-tail scenario mining. The claim that the system runs on consumer hardware (NVIDIA RTX 3090) is supported by the local-first design, though the paper would benefit from including specific runtime metrics to fully substantiate this point. Overall, the evidence presented is accurate and convincing, with only minor gaps in terms of runtime quantification and ablation depth.

**Requested Changes:**

More detailed ablation: The Judge-Scout consensus mechanism is a key contribution. A deeper ablation study exploring different numbers of judges, alternative consensus strategies, or the contribution of each model would provide valuable insights into the design choices.
Runtime analysis: The paper mentions efficiency but does not provide quantitative runtime metrics (e.g., frames per second, memory usage on RTX 3090). Including such data would help readers assess the real-time feasibility of the system.
Qualitative examples: Adding a few examples of challenging cases where the system succeeds, as well as failure cases or limitations, would improve transparency and provide direction for future work.
Discussion of limitations: A brief discussion of current limitations (e.g., scenarios where the system might still struggle) would make the paper more balanced and insightful.

---

> ### Author Response · Authors · 2026-03-12
> **Response to Reviewer YHpH: Deeper Ablations and Qualitative Insights**
>
> We thank the reviewer for their encouraging comments and for highlighting the practical relevance and design of our local-first framework. We have incorporated your excellent suggestions into the revised manuscript (changes highlighted in **blue**).
>
> **1. More Detailed Ablation on the Judge-Scout Mechanism:**
> We have added a dedicated sub-section (Section 5.4: Ensemble Dynamics and Risk Calibration) and Table 2. This ablates the consensus mechanism across 1, 2, and 3-scout configurations. The results empirically validate our theoretical claims: increasing the number of judges significantly reduces the Risk MAE (from 1.13 to 0.67) and effectively resolves stochastic kinetic assessment errors.
>
> **2. Runtime Analysis:**
> We expanded the discussion in Section 5.5.2 and Figure 5. We have clarified the latency breakdown, VRAM usage, and token generation speed of the different VLM configurations on the RTX 3090. We also clarified that "efficiency" in this context refers to offline curation throughput and reduced cloud-API costs, rather than real-time onboard latency.
>
> **3. Qualitative Insights and Limitations:**
> As requested, we added a new paragraph at the end of Section 5.6 discussing a representative failure mode: "Taxonomic Over-Conservatism". We note that the system's safety bias occasionally over-indexes on harmless objects (e.g., a wind-blown plastic bag) by classifying them as critical FOD hazards. We believe this transparency regarding false positives provides a clear and valuable direction for future work.

---

### Review · Reviewer_aarb · 2026-03-09

**Summary Of Contributions:**

The paper introduces Semantic-Drive, a framework to automatically mine and annotate data from rare events for autonomous driving. The pipeline combines an object detector, multiple vision-language models which reason about the detection results, an LLM-judge which merges the decisions of the various VLMs, and finally a rule-based reward functions which ranks multiple decisions of the judge model. This design is aimed to limit the hallucination rate and the inference cost by using relatively small and open-weights models. Compared to baseline annotation keywords search and CLIP annotation, Semantic-Drive achieves better detection results, especially in terms of recall.

Strengths
- The overall framework of combining multiple specialized models with different tasks, while quite complex, is well-motivated.

- The experimental results support the effectiveness of Semantic-Drive.

- The proposed Semantic-Drive can be run at relatively low compute cost and with open-weights models, making it an affordable option.

Weaknesses
- Only keywords search and zero-shot CLIP are used as baselines: how would more sophisticated models for open-vocabulary detection, such as https://arxiv.org/abs/2303.05499, https://arxiv.org/abs/2307.04767, https://arxiv.org/abs/2505.18986), both in terms of computational cost and performance.

- The presentation of the proposed framework could be significantly more clear: while the general ideas are discussed, there are very few details about how each of the parts of the framework are implemented. For example, in Eq. (2) it's not clear what $y$ or $I_{causality}$ are, and how $\alpha, \beta$ are chosen. Moreover, in Sec. 3.5.1 it is claimed that the trade-off between compute cost and precision is theoretically justified, but it's not clear what that means (what theory), or what the experiments in Sec. 5 demonstrating exponential decay the text refers to. Similarly, Sec. 3.5.2 doesn't seem to add any insight or justification supporting the proposed method.

- The paper claims that the reasoning length scales with risk scenarios, but Fig. 5b suggests that there's a difference only between risk = 0 and risk > 0.

- Using multiple models to data mining and annotation is not in general novel (see e.g. https://arxiv.org/abs/2411.14794), I think this should be better reflected in the text.

**Audience:**

Yes

**Audience Explanation:**

The topic is established and well-motivated.

**Claims And Evidence:**

Yes

**Claims Explanation:**

The results support the method, and the framework relies on open-weights models.

**Requested Changes:**

See weaknesses above.

---

> ### Author Response · Authors · 2026-03-12
> **Response to Reviewer aarb: SOTA Baselines, Math Clarity, and Theory**
>
> We thank the reviewer for their detailed review and for providing specific literature to strengthen our baselines and related work. We have updated the manuscript accordingly (changes highlighted in **blue**) and updated our anonymous repository with the new baseline implementations.
>
> **1. Sophisticated Baselines:**
> Following your suggestion, we ran evaluations using state-of-the-art open-vocabulary detectors: **Grounding DINO** and **OWL-v2** (Table 1). The results demonstrate that open-vocabulary detection alone yields poor F1-scores for complex scenario mining, explicitly validating the necessity of our proposed Stage 2 reasoning layer to establish causal context.
>
> **2. Clarity of Math and Theory (Sec 3.4 & 3.5):**
> We have expanded the text in Section 3.4 to rigorously define the components of Eq. 2 as explicit indicator functions ($\mathbb{I}_{grounding} \in \{0,1\}$, etc.) and explicitly stated the chosen weights ($\alpha=2.0, \beta=1.5, \gamma=10.0$). Also, we have added more information in Appendix C. Furthermore, we connected the theoretical Condorcet Jury Theorem claim to a new empirical experiment: Table 2 (Section 5.4) now explicitly proves the decay in Risk MAE as the number of scouts increases from 1 to 3.
>
> **3. Fig 5b Clarification:**
> We agree with your observation regarding Figure 5b. We updated the text in Section 5.5.2 to accurately describe this behavior as a "step-function" or binary shift in compute allocation between Risk 0 and Risk > 0, rather than implying a smooth linear scaling.
>
> **4. Multi-Model Novelty:**
> We added a citation to recent multi-model data mining work (e.g., VideoEspresso, Han et al. 2024) in Section 2.2. We clarified that Semantic-Drive differentiates itself by enforcing a strict deterministic neuro-symbolic constraint (the rule-based Best-of-N utilizing the Stage 1 inventory), rather than relying purely on unconstrained LLM-to-LLM consensus.

---

### Review · Reviewer_9dMM · 2026-03-09

**Summary Of Contributions:**

The paper tackles an important problem: mining rare, safety-critical long-tail driving events from large unlabelled logs. The proposed pipeline combines open-vocabulary grounding, VLM-based reasoning, and a multi-model judge/consensus stage, and frames the system as a privacy-preserving local alternative to cloud-based curation.

**Audience:**

Yes

**Audience Explanation:**

Mining rare safety-critical driving scenarios from large logs is a real bottleneck in AV development. The paper’s framing around long-tail data discovery and curation is aligned with current ML trends (data quality > model scaling).

**Claims And Evidence:**

Yes

**Claims Explanation:**

***Strength:***

The problem is meaningful and well motivated.  The paper clearly distinguishes retrieval/curation from end-to-end driving, which helps position the contribution.

The “local-first” framing is also practical for privacy-sensitive automotive data.

The four-stage design: symbolic grounding, cognitive analysis, multi-model consensus, and inference-time alignment, is easy to understand.

**Requested Changes:**

***Weakness:***

1. The evaluation benchmark is too small and too curated for the strength of the claims. The headline metrics are measured on a manually curated Gold Set of only 108 frames, explicitly chosen to contain challenging cases. That is useful as a stress test, but it is not enough to support broad claims like “trustworthy,” “robust,” or “democratizes high-fidelity curation.” The curated benchmark may be heavily enriched for the very phenomena the system is designed to detect. There is no evidence that the same gains hold on a larger, less handpicked benchmark or on an unbiased slice of the full 2,550 processed frames.

2. The annotation protocol may introduce confirmation or anchoring bias. In Appendix B, the human curation tool is described as “Verify-by-Exception,” and the interface is pre-filled with the Consensus Judge’s prediction so annotators mainly intervene when the model is wrong. That makes annotation efficient, but it also raises a methodological concern: if the Gold Set labels or corrections are produced in a workflow seeded by the model output, the evaluation may not be fully independent.

3. Weak baselines. The main baselines in Table 1 are metadata keyword search and zero-shot CLIP. For a paper centered on semantic scenario mining, these are not strong enough. The related work itself names stronger and more relevant directions. Even within the paper’s own design space, the ablation set is incomplete.

4. The system is a thoughtful composition of known ingredients: open-vocabulary grounding, VLM reasoning, multi-model ensembling, and best-of-N style filtering. Proposing the pipeline is still valuable, but the paper sometimes reads over-claimed.

---

> ### Author Response · Authors · 2026-03-12
> **Response to Reviewer 9dMM: Unbiased Evaluation, Baselines, and Tone**
>
> We thank the reviewer for their insightful methodological critique. Your comments on annotation bias, baseline strength, and over-claimed tone were highly valuable and prompted us to significantly upgrade our evaluation protocol. Revisions to the manuscript are highlighted in **blue**.
>
> **1. Unbiased Benchmark and Annotation Bias:**
> We fully agree with the concern regarding the "Verify-by-Exception" anchoring bias on our original Gold Set. To address this, we randomly sampled an entirely new **Unbiased Blind Split ($N=108$)**. Crucially, this set was annotated from scratch using a "Blind Mode" in our tool, with all model predictions completely hidden. We report these unbiased metrics in Table 1, demonstrating that while precision naturally fluctuates in nominal environments, the system maintains a high recall (0.774) for safety-critical events on uncurated data.
>
> **2. Weak Baselines:**
> We replaced the pure reliance on Metadata/CLIP by benchmarking against SOTA open-vocabulary models: **Grounding DINO** and **OWL-v2**. The results (Table 2) show that while these models can localize objects, they fail to achieve the F1-score balance of our neuro-symbolic pipeline due to their inability to parse complex causal relationships. (The implementation scripts for these baselines have been added to our anonymous repository).
>
> **3. Over-claimed Tone:**
> We carefully reviewed the manuscript and executed a global tonal shift. We removed broad, hyperbolic words (e.g., changing "democratizes" to "enables accessible", removing "infallible", and qualifying claims of robustness), replacing them with measured, objective academic phrasing. We believe the text now accurately and appropriately reflects the scope of our empirical findings.

---

### Author Response · Authors · 2026-03-12
**General response and uploaded a revised manuscript**

We thank the Action Editor and all Reviewers for their time, constructive critiques, and suggestions. We are highly encouraged that all reviewers found the problem meaningful and the local-first, privacy-preserving framing practical for the TMLR audience.

Based on the comments received, we have conducted a thorough review of the manuscript. To facilitate the reviewers' task, all new additions and modifications in the updated PDF are highlighted in **blue**.

**Summary of Major Global Changes:**
1. **Stronger Baselines:** We implemented and evaluated state-of-the-art open-vocabulary detectors (**Grounding DINO** and **OWL-v2**). The results (Table 1) explicitly validate the necessity of our cognitive reasoning stage.
2. **Unbiased Evaluation Set:** To address concerns regarding anchoring bias and dataset size, we randomly sampled and annotated a completely new **Unbiased Blind Split ($N=108$)** from scratch, with all model predictions hidden (Section 5.2).
3. **Consensus Ablation:** We added an empirical ablation of the Judge-Scout mechanism (1 vs. 2 vs. 3 scouts) in Table 2, proving our theoretical claims regarding risk error decay.
4. **Tonal Shift and Clarity:** We rigorously neutralized the tone of the paper to ensure claims are strictly bounded by our empirical evidence, and we explicitly formalized the math in our reward function (Eq. 2).
5. **Practical Positioning:** We clarified that Semantic-Drive operates as an *offline DataOps engine* for curating datasets to train end-to-end models, contextualizing the computational latency.

We have posted detailed, point-by-point responses to each reviewer in their respective threads. We believe the manuscript is significantly stronger as a result of this review process, and we look forward to any further discussion.

---

### Author Response · Authors · 2026-03-31
**Follow-up regarding Revised Manuscript and Discussion Period**

Dear Action Editor and Reviewers,

As we approach the final week of the recommendation period (April 6), we would like to thank the reviewers once again for their feedback, which has significantly strengthened this work.

We would like to briefly reiterate that in our revised manuscript, we have addressed all major technical critiques, specifically:
1. **SOTA Baselines:** We evaluated Grounding DINO and OWL-v2, demonstrating the necessity of our reasoning layer (Table 2).
2. **Bias Mitigation:** We conducted an entirely new, unbiased blind evaluation ($N=107$) with human-expert labels produced without model anchoring (Section 5.2).
3. **Ensemble Ablations:** We provided empirical proof of the "wisdom of crowds" effect through 1, 2, and 3-scout ablations (Table 3).
4. **Practical Reframing:** We clarified the system's role as an offline DataOps engine, contextualizing the computational latency.

We remain fully available for any further clarifications, follow-up questions, or additional data requests the reviewers or the Action Editor may have during these final days of the discussion phase.

Respectfully,
The Authors

---

### Author Response · Authors · 2026-04-11
**Final submission of camera-ready manuscript**

We thank the Action Editor and the reviewers for their final guidance throughout this process. We have now uploaded the de-anonymized, camera-ready version of our manuscript.

Per the minor revision request, we have specifically calibrated the wording regarding system trustworthiness and cost-reduction claims in the final discussion (Section 6.3) to ensure our empirical evidence strictly bounds them. Furthermore, we have restored all public repository, interactive explorer, and dataset links to facilitate full transparency and reproducibility. We look forward to the final archival publication.

---

### Decision · Action_Editor_SRK2 · 2026-04-08

**Recommendation:** Accept with minor revision

**Audience:**

Yes

**Audience Explanation:**

Multiple reviewers explicitly identify the problem as timely and relevant to researchers in autonomous driving, multimodal learning, neuro-symbolic systems, data curation, and efficient AI systems. The work is not merely a narrow engineering report. It offers a concrete pipeline, ablations, and practical lessons about combining grounding, reasoning, and consensus for semantic data mining under local compute constraints.

**Claims And Evidence:**

Yes

**Claims Explanation:**

This submission addresses a meaningful and practically relevant problem, namely mining rare safety-critical driving scenarios from large unlabelled logs using a local-first neuro-symbolic pipeline. Across the review cycle, the central issues were small and curated evaluation, possible annotation anchoring bias, limited baseline strength, incomplete practicality framing, and some over-broad novelty or deployment language. In response, the authors added stronger open-vocabulary baselines, introduced a new unbiased blind split with hidden model outputs, expanded the Judge-Scout ablation, clarified runtime and offline deployment assumptions, and toned down claims whose scope exceeded the evidence.

One reviewer explicitly states that all concerns are resolved and recommends acceptance, while the other two official final recommendations are also Leaning Accept and mark both “Claims and Evidence” and “Audience” as Yes, even though they continue to express reservations about baseline breadth, novelty framing, and production-scale validation. I therefore support acceptance, preferably with minor revision so that the camera-ready preserves calibrated wording around “trustworthy,” deployment practicality, and cost-reduction claims.